**Technical Note: Evaluation of simultaneous measurements of mesospheric OH, HO$_2$, and O$_3$ under photochemical equilibrium assumption: Statistical approach**

Mikhail Yu. Kulikov[1], Anton A. Nechaev[1], Mikhail V. Belikovich[1], Tatiana S. Ermakova[1], and Alexander M. Feigin[1]

[1]Institute of Applied Physics of the Russian Academy of Sciences, 46 Ulyanov Str., 603950 Nizhny Novgorod, Russia

Correspondence to: Mikhail Yu. Kulikov (mikhail_kulikov@mail.ru)

**Abstract**

The Technical Note presents a statistical approach to evaluating simultaneous measurements of several atmospheric components under the assumption of photochemical equilibrium. We consider simultaneous measurements of OH, HO$_2$, and O$_3$ at the altitudes of the mesosphere as a specific example and their daytime photochemical equilibrium as an evaluating relationship. A simplified algebraic equation relating local concentrations of these components in the 50-100 km altitude range has been derived. The parameters of the equation are temperature, neutral density, local zenith angle, and the rates of 8 reactions. We have performed a one-year simulation of the mesosphere and lower thermosphere using a 3D chemical-transport model. The simulation shows that the discrepancy between the calculated evolution of the components and the equilibrium value given by the equation does not exceed 3-4% in the full range of altitudes independent of season or latitude. We have developed the technique of statistic Bayesian evaluation of simultaneous measurements of OH, HO$_2$ and O$_3$ based on the equilibrium equation taking into account the measurement error. The first results of application of the technique to MLS/Aura data are presented in this Technical Note. It has been found that the satellite data of HO$_2$ distribution regularly demonstrates essentially lower altitudes of mesospheric maximum of this component. This has also been confirmed by model HO$_2$ distributions and comparison with offline retrieval of HO$_2$ from the daily zonal means MLS radiance.

## 1. Introduction

A prominent feature of the atmospheric photochemical systems is the presence of a large number of chemical components with short lifetime and concentrations close to stable photochemical equilibrium at every instant. The condition of balance between their sources and sinks is described by a system of algebraic equations. This system can be used to determine characteristics of hard to measure atmospheric species through other measurable components, validate results of remote or *in situ* measurements, estimate reaction rates usually known with significant uncertainty, and to understand processes and chemical reactions that influence variability of the most important atmospheric components, e.g. ozone, in the geographical region of interest.

This approach has found wide application:

(1) in 3D chemical transport models that include a large set of physical and chemical processes with a broad spectrum of spatio-temporal scales. In particular, the chemical family concept is widely used for simulating gas phase photochemistry of the lower and middle atmosphere (e.g., Douglass et al., 1989; Kaye and Rood, 1989; Rasch et al., 1995), when transport is taken into account only for the concentration of a chemical family, while relative concentrations of the constituent fast components are calculated from the instantaneous stable equilibrium condition. Complemented with the Henry law (e.g., Djouad et al., 2003; Tulet et al., 2006) in multiphase models, this approach markedly saves calculation time and increases the overall stability of the numerical scheme. Moreover, the use of the photochemical equilibrium condition to simulate fast components dynamics reduces the phase space dimension of box models significantly (e.g., Kulikov and Feigin, 2014), allowing a comprehensive analysis of nontrivial nonlinear dynamic properties of various atmospheric photochemical systems (e.g., Feigin and Konovalov, 1996; Feigin et al., 1998; Konovalov et al., 1999; Konovalov and Feigin, 2000; Kulikov et al., 2012).

(2) in investigations of the chemistry of the surface layer and free troposphere in different regions (over megalopolises, in rural areas, in the mountains, over the seas) based on measurements of nitrogen species, peroxy radicals, ozone, aerosols, and other components aimed at understanding processes impacting the surface ozone formation and air quality. The equilibrium condition is most frequently used for nitrogen species. For example, Chameides (1975) proposed a model for determining the vertical distribution of odd nitrogen, in which the $HNO_3$ profile could be

deployed to retrieve profiles of five other components (NO, $NO_2$, $NO_3$, $N_2O_5$, and $HNO_2$) from their
photochemical equilibrium condition. In the paper by Stedman et al. (1975) the equation for $NO_2$
equilibrium that accounted only for the main source and sink of this component was applied to
determine the photodissociation constant $J(NO_2)$. A more accurate equation for the $NO_2$ equilibrium
was used by Crawford et al. (1996) and Kondo et al. (1996) to determine the $NO_2$/NO partitioning
and $NO_x$, allowing, in particular, investigating the spatial distribution of $NO_x$/$NO_y$ over the Pacific.
Night-time equilibrium in the $NO_2$-$NO_3$-$N_2O_5$ system is used to determine surface layer $N_2O_5$
concentration, equilibrium constant of this system, equilibrium partitioning between $NO_3$ and $N_2O_5$,
and loss coefficients of $NO_3$, $N_2O_5$ and $NO_x$ (Martinez et al., 2000; Brown et al., 2003; Crowley et
al., 2010; McLaren et al., 2010; Benton et al., 2010; Sobanski et al., 2016).
Platt et al. (1979) used the $CH_2O$ photochemical equilibrium condition to analyse results of
simultaneous measurement of $CH_2O$, $O_3$ and $NO_2$ and to identify mechanisms of $CH_2O$ formation
over rural areas and in maritime air. In the papers by Ko et al. (2003), Cantrell et al. (2003),
Penkett et al. (1997), Penkett et al. (1998) algebraic expressions derived from equilibrium
conditions for $H_2O_2$, peroxy radicals and nitrogen species were used to determine equilibrium
values of peroxide concentration, total peroxy radical level, and NO/$NO_2$ ratio, and to diagnose the
ozone production and loss levels in clean or polluted troposphere.
(3) in stratospheric chemistry studies, including determination of a critical parameter in
catalytic cycles of ozone destruction in the polar stratosphere. In particular, the equilibrium
condition for ClO and $Cl_2O_2$ along with the measurement data of daytime and night-time
concentrations of these components in the polar stratosphere are used to evaluate the temperature
dependence of the ClO concentration, reaction constants determining the
$ClO + ClO + M \leftrightarrow Cl_2O_2 + M$ equilibrium, and the photolysis rate of $Cl_2O_2$ (Ghosh et al., 1997;
Avallone et al., 2001, Solomon et al., 2002; Stimpfle et al., 2004; von Hobe et al., 2005; Berthet et
al., 2005; Butz et al., 2007; von Hobe et al., 2007; Kremser et al., 2011; Sumińska-Ebersoldt et al.,
2012; Wetzel et al., 2012).
Pyle et al. (1983) proposed a method for derivation of the OH concentration from satellite
infrared measurements of $NO_2$ and $HNO_3$ using a simple algebraic relation following from the
equilibrium condition for $HNO_3$. Algorithms for retrieving distributions of OH and HO2 from the
satellite measurement data of $O_3$, $NO_2$, $H_2O$, $HNO_3$ by LIMS/Nimbus 7 and UARS with the help of
algebraic models following from the photochemical equilibrium of $O_x$, $HO_x$ and $HNO_3$ components
were proposed by Pyle and Zavody (1985), Pickett and Peterson (1996). It is also worthy of note
that similar models are widely used for calculating concentrations of components with a short
lifetime (e.g. $O(^1D)$ and OH) and subsequent evaluating vertical distributions of eddy diffusivity from
measurements of trace gas concentration profiles (see, e.g., Massie and Hunten, 1981).
Kondo et al. (1988) made use of the photochemical equilibrium between NO and $NO_2$ for
understanding diurnal variations of NO concentration measured during aircraft flights. In the paper
by Webster et al. (1990) simultaneous *in situ* balloon-borne measurements of NO, $NO_2$, $HNO_3$, $O_3$
and $N_2O$ and the photochemical equilibrium condition for various nitrogen components were used
to determine OH, $N_2O_5$ and $NO_y$ concentrations. A similar approach was employed by Kawa et al.
(1990), who obtained $NO_2$, $N_2O_5$, $ClNO_3$, $HNO_3$ and OH concentrations from aircraft measurements
of NO, ClO and $O_3$ concentrations. Hauchecorne et al. (2010) found that $NO_3$ concentration
measured by GOMOS/ENVISAT positively correlates with temperature at altitudes up to 45 km in
the region where $NO_3$ is in chemical equilibrium with $O_3$. Funke et al. (2005) used NO and $NO_2$
stable-state photochemistry to verify correctness of the new approach of retrieving distributions of
those component from MIPAS/ENVISAT measurement data. Marchland et al. (2007) proposed a
method to retrieve the temperature distribution in the stratosphere between 30 km and 40 km from
$O_3$ and $NO_3$ measurements by GOMOS with the help of a simple equation derived from the night-
time $NO_3$ chemical equilibrium.
(4) in investigations of the chemistry of $O_x$–$HO_x$ components and atmospheric glows in the
mesosphere and MLT area. In particular, Kulikov et al. (2006, 2009) proposed algorithms for the
simultaneous retrieval of O, H, $HO_2$ and $H_2O$ from joint OH and $O_3$ satellite measurement, in which
the assumption of photochemical equilibrium of $O_3$, OH, and $HO_2$ was utilized. For several decades
the assumption of the photochemical equilibrium of ozone (PEO) was widely used to determine
distributions of atomic oxygen and atomic hydrogen at altitudes of the MLT via satellite and rocket
measurement of ozone concentration and airglow emissions (e.g., Evans and Llewellyn, 1973;
Good, 1976; Pendleton et al., 1983; McDade et al., 1985; McDade and Llewellyn, 1988; Evans et
al., 1988; Thomas, 1990; Llewellyn et al., 1993; Llewellyn and McDade, 1996; Mlynczak et al.,
2007, 2013a, 2013b, 2014; Smith et al., 2010; Siskind et al., 2008, 2015). Russell and Lowe (2003)
applied PEO to infer the seasonal and global climatology of atomic oxygen using WINDII/UARS.
PEO was deployed to investigate hydroxyl emission mechanisms, morphology, and variability in
the upper mesosphere – lower thermosphere region (Marsh et al., 2006; Xu et al., 2010, 2012;

Kowalewski et al., 2014). Mlynczak and Solomon (1991, 1993) and Mlynczak et al. (2013b) used the equilibrium assumption to derive exothermic chemical heat. The PEO assumption employed for studying the mesospheric OH* layer response to gravity waves (Swenson and Gardner, 1998). In ultimately theoretical works, e.g. Grygalashvyly et al. (2014), Grygalashvyly (2015), PEO was used to derive the dependence of excited hydroxyl layer concentration and altitude on atomic oxygen and temperature. In the paper by Sonnemann et al. (2015) it was used to analyze annual variations of OH* layer. Moreover, PEO is frequently applied implicitly, when authors are equating the night-time loss of ozone in the reaction with atomic hydrogen and production of ozone by a 3-body reaction of molecular and atomic oxygen (e.g., Nikoukar et al., 2007).

In the present Technical note we demonstrate how the photochemical equilibrium condition of several atmospheric components may be employed to statistically validate data of their simultaneous measurements, particularly in the case when measurement error is large.

We consider the simultaneous photochemical daytime equilibrium of OH, $HO_2$, and $O_3$ at the altitudes of the mesosphere. We have derived a simplified algebraic equation

$$F(OH, HO_2, O_3) = 1,$$

describing the relationship between local concentrations of the components at the altitudes of 50–100 km. The only parameters of the equation are temperature, neutral density, local zenith angle, and constants of 8 reactions. One-year simulation of the mesosphere and lower thermosphere based on a 3D chemical-transport model shows that the discrepancy between the calculated evolution of the components and the equilibrium value given by the equation does not exceed 3–4 % in the full range of altitudes independent of season or latitude.

We have developed a technique of statistical Bayesian evaluation of simultaneous measurement of OH, $HO_2$ and $O_3$ based on the mentioned equilibrium equation taking into account the measurement error. The first results of its application to MLS/Aura data (Wang et al., 2015a,b; Schwartz et al., 2015) are presented. It is found that the satellite data of $HO_2$ distribution regularly demonstrates essentially lower altitudes of this component's mesospheric maximum. These results confirm the ones obtained via the offline retrieval of $HO_2$ from the MLS primary data (Millán et al., 2015).

The Technical Note is structured as follows. A 3D chemical transport model is briefly described in Sect. 2. In Sect. 3 a simplified algebraic relationship between the equilibrium concentrations of OH, $HO_2$ and $O_3$ is derived and verified by 3D simulations. Section 4 presents the

method of statistical evaluation of simultaneous data of OH, HO$_2$ and O$_3$. The results of applying
the method to MLS/Aura data are presented in Sect. 5. The last Section contains discussion of the
results followed by concluding remarks.

**2. Model and calculations**

For our calculations we used the global 3D chemical transport model (CTM) of the middle
atmosphere developed by the Leibniz Institute of Atmospheric Physics (IAP) (e.g., Sonnemann et
al., 1998). It was designed particularly for investigation of the spatio-temporal structure of
phenomena in the MLT region and specifically in the extended mesopause region. The grid-point
model extends from the ground up to the middle thermosphere (0–150 km; 118 pressure-height
levels). The horizontal resolution amounts to 5.625° latitudinally and 5.625° longitudinally. The
chemical module described in numerous papers (e.g., Sonnemann et al., 1998; Körner and
Sonnemann, 2001; Grygalashvyly et al., 2009, 2011, 2012) consists of 19 constituents, 49
chemical reactions, and 14 photo-dissociation reactions (see Table 1). The reaction rates used in
the model are taken from Burkholder et al. (2015). The temperature-dependent reaction rates are
calculated on-line, thus, they are sensitive to small temperature fluctuations. We make use of the
pre-calculated dissociation rates (Kremp et al., 1999).
The evolution of the components of HO$_x$ (H, OH, HO$_2$, H$_2$O$_2$) and NO$_x$ (N, NO, NO$_2$, NO$_3$)
families is calculated using the chemical family concept proposed by Shimazaki (Shimazaki, 1985).
This is done because of the presence of short-lived components among these families, with
lifetimes much shorter than those of the families themselves, which imposes significant restrictions
on the value of the CTM's integration step. For example, the daytime lifetimes of OH and HO$_2$
above 70 km are about 1 s or less, while the lifetime of the HO$_x$ family is about $10^4$ s or more.
Therefore, when calculating these components individually it is necessary to set the CTM's
integration step to be much less than 1 s. In our work, the Shimazaki technique is applied for
calculating the evolution of each component of the HO$_x$ and NO$_x$ families. We emphasize that this
technique does not explicitly use the steady-state approximation for the components, instead it
utilizes the approach based on an implicit Euler scheme (see Shimazaki, 1985). This allows
increasing the integration step of CTM significantly without loss of accuracy of calculating the short-
lived components. In our work the integration time is chosen to be 9 s.

The model includes 3D advective and vertical diffusive transport (turbulent and molecular). Three-dimensional fields of temperature and winds are taken from the Canadian Middle Atmosphere Model (CMAM) for the year 2000 (de Grandpre et al., 2000; Scinocca et al., 2008). We use the Walcek-scheme (Walcek and Aleksic, 1998; Walcek, 2000) for advective transport and the implicit Thomas algorithm as described in Morton and Mayers (1994) for diffusive transport. The vertical eddy diffusion coefficient is based on the results by Lübken (1997).

The CTM driven by COMMA-IAP middle atmosphere dynamics (Berger, 1994; Ebel et al., 1995; Kremp et al., 1999; Berger and von Zahn, 1999) was verified by measurements, particularly for ozone, in a number of papers (Hartogh et al., 2004, 2011; Sonnemann et al., 2006, 2007).

We calculate the annual variation of spatio-temporal distributions of OH, $HO_2$, and $O_3$ and constructed distributions of the $F(OH, HO_2, O_3)$ function introduced in Sect. 1. To remove transitional regions that correspond to sunset and sunrise, we take into account only periods of local time with the solar zenith angle $\chi < 85°$. The obtained results are presented in the model coordinates, so the pressure-height levels are used for the vertical axes. In addition, the approximate altitudes are shown in the figures of Sec. 1, calculated for a given month utilizing averaged temperature profiles of the model and hydrostatic equilibrium.

## 3. Daytime photochemical equilibrium of OH, $HO_2$, and $O_3$ at the altitudes of the mesosphere

The daytime balance of OH concentration at mesospheric altitudes is determined by the following primary reactions (Brasseur and Solomon, 2005):

$HO_2 + O \rightarrow OH + O_2$ (R18 in Table 1)

$H + O_3 \rightarrow OH + O_2$ (R21)

$H + HO_2 \rightarrow 2OH$ (R14)

$OH + O \rightarrow H + O_2$ (R17)

$OH + O_3 \rightarrow HO_2 + O_2$ (R22)

The daytime balance of $HO_2$ concentration:

$H + O_2 + M \rightarrow HO_2 + M$, M is molecule of air (R20)

$OH + O_3 \rightarrow HO_2 + O_2$ (R22)

$HO_2 + O \rightarrow O_2 + OH$ (R18)

The daytime balance of $O_3$ concentration:

$O + O_2 + M \rightarrow O_3 + M$ (R12)
$O_3 + h\nu \rightarrow O_2 + O$ (R52)
$O_3 + h\nu \rightarrow O_2 + O(^1D)$ (R53)
$O_3 + H \rightarrow OH + O_2$ (R21)

Expressions for local concentrations of OH, $HO_2$, and $O_3$ in the photochemical equilibrium

are written in the form
$$OH = \frac{k_{18} \cdot HO_2 \cdot O + 2k_{14} \cdot HO_2 \cdot H + k_{21} \cdot O_3 \cdot H}{k_{17} \cdot O + k_{22} \cdot O_3} , \qquad (1)$$

$$HO_2 = \frac{k_{20} \cdot M \cdot O_2 \cdot H + k_{22} \cdot O_3 \cdot OH}{k_{18} \cdot O} , \qquad (2)$$

$$O_3 = \frac{k_{12} \cdot M \cdot O_2 \cdot O}{k_{52} + k_{53} + k_{21} \cdot H} , \qquad (3)$$

where $k_i$ are the corresponding reaction constants from Burkholder et al. (2015).
We eliminate O and H from Eqs. (1)-(3) and derive an expression depending only on OH, $HO_2$, $O_3$.

Almost everywhere in the mesosphere and lower thermosphere (with the exception of 85-95

km, see Kulikov et al., 2017) the photodissociation is the main ozone sink, i.e. $k_{52} + k_{53} >> k_{21} \cdot H$.
Therefore, in the zero order approximation Eq. (3) can be simplified and the concentration of
atomic oxygen can be defined in terms of ozone concentration:
$$O = \frac{k_{52} + k_{53}}{k_{12} \cdot M \cdot O_2} O_3 \qquad (4)$$

Making use of Eq. (4) we can derive from Eq. (2) an expression for the concentration of H in terms
of concentrations of OH, $HO_2$ and $O_3$:
$$H = \frac{k_{18} \cdot (k_{52} + k_{53}) / (k_{12} \cdot M \cdot O_2) \cdot HO_2 - k_{22} \cdot OH}{k_{20} \cdot M \cdot O_2} O_3 \qquad (5)$$

By substituting this equation and Eq. (4) into Eq. (1) we obtain an expression relating OH, $HO_2$, and
$O_3$:
$$F(OH, HO_2, O_3) = \left( \frac{k_{20} \cdot M \cdot O_2}{k_{20} \cdot M \cdot O_2 + k_{21} \cdot O_3 + 2 \cdot k_{14} \cdot HO_2} + \frac{k_{12} \cdot M \cdot O_2 \cdot k_{22}}{(k_{52} + k_{53}) \cdot k_{17}} \right) \cdot \frac{k_{17} \cdot OH}{k_{18} \cdot HO_2} = 1 \qquad (6)$$

Figure 1 shows height–latitude cross-sections of $< F(OH, HO_2, O_3) >$ for each month (in this

Section angle brackets denote monthly averaged zonal mean values). The dashed area
corresponds to $\chi > 85°$. One can see that eq. (15) is most accurate within the 50–76 km range and
above 86 km, where $|<F> - 1| \leq 1\%$. The difference reaches 3–4 % in the region between 76 km
and 86 km. The altitude of this region has an annual variation with a maximum deviation in the
winter hemisphere. Below 50 km the value of $<F>$ increases up to 1.2 at 40 km, thus below the
stratopause Eq. (6) no longer describes the simultaneous photochemical equilibrium of OH, $HO_2$
and $O_3$. Note that these components remain short-lived below 50 km (with the lifetimes of about
$10^2$-$10^3$ s (Brasseur and Solomon, 2005)) depending on height and duration of daylight. However,
for quantitative description of their daytime equilibrium it is necessary to include additional
reactions involving, in particular, the components of the $NO_x$ family.

Note also that Eq. (1) and Eq. (6) take into account only the main daytime source of OH

($P_{OH}$) specified by reactions R18, R14, and R21:
$P_{OH} = k_{18} \cdot HO_2 \cdot O + 2k_{14} \cdot HO_2 \cdot H + k_{21} \cdot O_3 \cdot H$
These reactions run "inside" the HOx (H, OH, $HO_2$, $H_2O_2$) family and do not perturb its total
concentration. The height–latitude cross-sections of $<P_{OH}>$ for each month are presented in
Fig. 2.
The next important daytime source of OH is specified by reactions R59 and R7 involving $H_2O$, the
main source for the $HO_x$ family:
$P_{OH}{}^{H_2O} = (k_{59} + 2 \cdot k_7 \cdot O(^1D)) \cdot H_2O$
Figure 3 shows height–latitude cross-sections of $<P_{OH}{}^{H_2O} / P_{OH}>$ for each month. Comparing Fig. 1
and Fig. 3, we conclude that the previously indicated 3–4 % deviation of $<F>$ from 1 in the region
between 76 km and 86 km is largely due to the neglect of these reactions.

Another source of OH is sporadically activated during charged particle precipitation events

and exists for a relatively short time (several days). Solar proton events (SPE) perturb the ionic
composition in the mesosphere and the upper stratosphere considerably and trigger a whole
cascade of reactions involving ions, neutral components and their clusters (e.g., $O_2^+ \cdot H_2O$). This
leads to an additional (to reactions R59 and R7) conversion of $H_2O$ molecules into OH and H
(Solomon et al., 1981). The maximum of the OH production rate ($P_{OH}{}^{SPE}$) induced by SPE is
located in the polar latitudes in the region of 60–80 km and, as a rule, does not exceed $2 \cdot 10^3$ cm$^{-3}$
s$^{-1}$ (Jackman et al., 2011, 2014). It can be seen from Fig. 2 that at these latitudes and altitudes the
$P_{OH}{}^{SPE} / P_{OH}$ ratio does not exceed 1-2%, even for the maximum values of $P_{OH}{}^{SPE}$. This means that
the impact of $P_{OH}^{SPE}$ on Eq. (6) is of the same order of smallness as in the case of reactions R59
and R7, hence, it may be neglected. A similar conclusion can be made for other reactions from
Table 1, not accounted for by Eq. (6), including the ones involving $NO_x$ in both quiet and perturbed
conditions in the mesosphere.

**4. Method of statistical evaluation of simultaneous measurement of OH, HO$_2$ and O$_3$**

The proposed method is based on the statistical Bayesian procedure described in the works by
Kulikov et al. (2009) and Nechaev et al. (2016). It was originally developed for retrieving trace gas
concentrations in the mesosphere from ground-based and satellite measurements of other
mesospheric components. With respect to the considered evaluation problem this procedure
consists of three steps: (1) constructing conditional probability density function (PDF) of OH, HO$_2$
and O$_3$ concentration values at each altitude *z* in the selected interval assuming that there is
certain measurement data of these components and the algebraic relationship (6) is valid; (2)
calculating the first moments of this distribution, i.e. expected value and dispersion of each
component using the Metropolis-Hastings algorithm (Chib and Greenberg, 1995) for
multidimensional integration; (3) comparing the obtained results with the initial measurement data.
For constructing posterior PDF it is convenient to introduce vector $\vec{u}\left\{HO_2^{ret}, O_3^{ret}, OH^{ret}\right\}$,
whose components are the retrieved values of chemical species concentrations at a certain altitude
*z*, and vector $\vec{x}\left\{HO_2^m, O_3^m, OH^m\right\}$ composed of experimentally measured values of the components
of vector $\vec{u}$, $x_j = u_j + \xi_j$, $j = 1..3$, where $\xi_j$ is a random error of measuring the *j*-th component of
vector $\vec{u}$ at the altitude *z*. It is assumed that
(1) random variables $\xi_j$ are distributed normally with densities
$$w_j(\xi_j) = \frac{1}{\sigma_j \sqrt{2\pi}} exp\left(-\frac{\xi_j^2}{2\sigma_j^2}\right);$$
(7)

(2) $\xi_j$ are mutually independent:
$$\vec{\xi}\left\{\xi_1, \xi_2, \xi_3\right\} \sim W_\xi(\vec{\xi}) = \prod_j w_j(\xi_j),$$
(8)

where $W_\xi(\vec{\xi})$ is the total PDF of all $\xi_j$;
(3) dispersions $\sigma_j$ in Eq. (7), that are expected error values, are assumed to be known a priori (in
our case they are provided by the MLS retrieval algorithm along with measured data).
Then the probability to observe vector $\vec{x}$ is given by the conditional PDF
$P_x(\vec{x}\,|\,\vec{u}) = \int \delta(\vec{x} - \vec{u}) W_\xi(\vec{\xi}')d^3\vec{\xi}' = W_\xi(\vec{x} - \vec{u})$,    (9)
where $\delta(\ldots)$ is delta function.

The prior relationship of $HO_2^{ret}$, $O_3^{ret}$ and $OH^{ret}$ concentrations (Eq. (6)) can be written as

$u_3 = G(u_1, u_2)$. Integrating the left-hand side of Eq. (17) with conditional PDF of the variable $u_3$:
$P_{u_3}(u_3\,|\,u_1,u_2) = \delta(u_3 - G(u_1,u_2))$,
yields a likelihood function of the model
$P_x(\vec{x}\,|\,u_1,u_2) = w_3\big(x_3 - G(u_1,u_2)\big)\cdot w_1(x_1 - u_1)w_2(x_2 - u_2)$.    (10)
According to Bayes' theorem, the posterior function, i.e. the probability density of latent variables $u_1$
and $u_2$, under the condition that $\vec{x}$ is observed, is defined by the expression
$$P(u_1,u_2\,|\,\vec{x}) \propto P_x(\vec{x}\,|\,u_1,u_2)\cdot P_{apr}(u_1,u_2)$$
$$\propto exp\left(-\frac{(x_1 - u_1)^2}{2\sigma_1^2}\right)\cdot exp\left(-\frac{(x_2 - u_2)^2}{2\sigma_2^2}\right)\cdot exp\left(-\frac{(x_3 - G(u_1,u_2))^2}{2\sigma_3^2}\right)\cdot P_{apr}(u_1,u_2)$$
   (11)

in which $P_{apr}(u_1,u_2)$ defines prior PDF of $u_1$ and $u_2$.

The retrieved value of the latent variable $u_{1,2,3}$ is hereinafter understood as the mean value

of the function in Eq. (11):
$$\langle u_{1,2}\rangle = \int_{-\infty}^{\infty}\int_{-\infty}^{\infty} u_{1,2}\cdot P(u_1,u_2\,|\,\vec{x})du_1 du_2$$
$$\langle u_3\rangle = \int_{-\infty}^{\infty}\int_{-\infty}^{\infty} G(u_1,u_2)\cdot P(u_1,u_2\,|\,\vec{x})du_1 du_2.$$
   (12)

Its dispersion defines the uncertainty of the retrieval:
$\sigma_{u_j} = \sqrt{\langle u_j^2\rangle - \langle u_j\rangle^2}$, $j = 1.3$,    (13)
where the angle brackets denote averaging in the sense of Eq. (12).

**5. MLS/Aura data evaluation and results**

We used the latest version (v4.2) of the MLS "standard" product (Livesey et al., 2017) for trace gas concentrations and temperature $T$ within the $1-0.046$ mbar pressure interval where all data are suitable for scientific applications (Wang et al., 2015a,b; Schwartz et al., 2015). We took the daytime data when the solar zenith angle $\chi < 80°$ for January, May, and September 2005. All data were appropriately screened. "Pressure", "estimated precision", "status flag", "quality", "convergence" and "clouds" fields were taken into account. $HO_2$ data were seen as the day-minus-night difference as prescribed by the MLS data guidelines (Livesey et al., 2017). Following Pickett et al. (2008), each daytime profile of this component measured on a given day at a latitude Lat, a profile resulting from averaging the nighttime profiles of $HO_2$, measured on the same day in the latitude range of Lat±5°, was subtracted. This operation eliminates systematic biases affecting $HO_2$ retrievals, but limits the studied latitude range to the one where MLS observes both daytime and nighttime data.

The integrals in Eq. (12)–(13) were calculated at every pressure level $p$ for each set of simultaneously measured vertical profiles $OH^{MLS}(p)$, $HO_2^{MLS}(p)$, $O_3^{MLS}(p)$, $T^{MLS}(p)$, $\sigma_{OH^{MLS}}(p)$, $\sigma_{HO_2^{MLS}}(p)$, $\sigma_{O_3^{MLS}}(p)$. The vertical profiles $<OH^{ret}>(p)$, $<HO_2^{ret}>(p)$, $<O_3^{ret}>(p)$, $\sigma_{OH^{ret}}(p)$, $\sigma_{HO_2^{ret}}(p)$, $\sigma_{O_3^{ret}}(p)$ were found at each point of the globe along the satellite track. Numerical integration was performed by a Monte Carlo method. For each pressure level, a sample of about $5 \cdot 10^5$ pairs of random variable values $\{u_1, u_2\} = \{HO_2^{ret}, O_3^{ret}\}$ distributed with normalized probability density given by Eq. (11) with $P_{apr}(u_1, u_2) \equiv 1$ was generated with the help of the Metropolis-Hastings algorithm (Chib and Greenberg, 1995). In this case, the statistical moments in Eq. (12)–(13) were determined by summation over the sample.

A typical example of retrieved profiles $HO_2^{ret}$, $O_3^{ret}$ and $OH^{ret}$ (black curves) in comparison with the measured $HO_2^{MLS}$, $O_3^{MLS}$ and $OH^{MLS}$ (red curves) is given in Fig. 4. First of all, note that statistics of the retrieved data is in satisfactory agreement with the initial measurement of OH and $O_3$ concentrations, but not of $HO_2$. The error of satellite measurement, $\sigma_{HO_2^{MLS}}$, greatly exceeds the uncertainty of retrieval, $\sigma_{HO_2^{ret}}$, so at some altitudes the values of $<HO_2^{MLS}>$ (red dashed curves) do not fall within the corresponding intervals $<HO_2^{ret}>\pm\sigma_{HO_2^{ret}}$. Second, the results of a single

measurement of all three components and their retrieved values have considerable uncertainties relative to their means within the whole interval of altitudes. Therefore, the observed and retrieved data should be compared using the commonly accepted approach (e.g., Pickett et al., 2008) of averaging large ensembles of profiles within certain latitude and time ranges, or zones. It is supposed that the noise of satellite measurement instruments is delta-correlated, so that random values corresponding to each single measured or retrieved profile are statistically independent. In this case the dispersion of a measured or retrieved zonal mean profile is determined by summation

$$\sigma^2_{\Sigma} = \frac{1}{N^2} \sum_{k=1}^{N} \sigma^2_{k} ,$$

where $N$ is the number of measured or retrieved profiles within the zone and $\sigma^2_{k}$ is the dispersion of the $k$-th measured or retrieved profile.

The range of latitudes covered by the satellite trajectory was divided into 17 bins $10^0$ each. About 3000 single profiles of each chemical component fall into one bin during a month of MLS/Aura observations. Therefore, the resulting uncertainties due to measurement noise of OH, $HO_2$ and $O_3$ concentration profiles (both measured and retrieved) averaged over such ensembles are significantly (about one and a half order of magnitude) lower than the uncertainties of individual profiles. Examples of such profiles for January, May and September 2005 are presented in Fig. 5. One can see that the indicated uncertainties are now small enough to make clear conclusions about the extent to which the observed and retrieved profiles agree by comparing their averaged values only, i.e. $< OH^{MLS} >$, $< HO_2^{MLS} >$, $< O_3^{MLS} >$ and $< OH^{ret} >$, $< HO_2^{ret} >$, $< O_3^{ret} >$.

Figures 4–6 show monthly averaged zonal mean pressure–latitude cross-sections of $< HO_2^{ret} >$, $< HO_2^{MLS} >$, $\Delta HO_2 = (< HO_2^{ret} > - < HO_2^{MLS} >)/ < HO_2^{MLS} >$ and similar characteristics for OH and $O_3$ concentration profiles for three months of the year 2005. First, clearly, the distributions of $< OH^{ret} >$ and $< O_3^{ret} >$ are in good qualitative and quantitative agreement with the initial MLS/Aura measurement data at lower altitudes, below $\sim 0.07$ mbar and $0.1$ mbar, correspondingly. At higher altitudes, the distributions of $< OH^{ret} >$ reproduce all the main structural features of $< OH^{MLS} >$, but the retrieved OH concentration has lower values than the observed one with a relative difference $\Delta OH$ reaching ~15% at the top. The distribution of $< O_3^{ret} >$ above $0.1$ mbar, in turn, differs considerably from $< O_3^{MLS} >$, both in quantity and quality, and $\Delta O_3$ locally

reaches 50-60% and more. Second, for all months there are significant qualitative and quantitative
differences between $<HO_2^{ret}>$ and $<HO_2^{MLS}>$, the most noticeable one being location of the
mesospheric maximum of this component's concentration. According to the observations it is close
to 0.1 mbar, while the retrieved data demonstrate the altitudes of about ~0.046 mbar or higher. Our
analysis of the applied method of statistical evaluation demonstrates that the higher position of this
maximum in the distributions of $<HO_2^{ret}>$ is influenced by the $OH^{MLS}$ data in which the
mesospheric maximum (see Figs. 6-8) is also located notably higher than 0.1 mbar.

**6. Discussion and conclusion**

On the basis of the data presented in Section 5 we can conclude that, upon the whole,
simultaneous OH, $HO_2$ and $O_3$ satellite measurements poorly satisfy the photochemical equilibrium
condition. The $HO_2$ component biases from this condition most prominently. We can conjecture that
a possible explanation for the bias is the significant systematic error in $HO_2$ measurements, in
particular, in the height of the mesospheric maximum. This assumption is supported by the
calculation of the $HO_2$ distributions with the use of our 3D chemical transport model (see Fig. 9). It
can be seen that the mesospheric maximum of $HO_2$ in these months, as well as of the $<HO_2^{ret}>$
distributions, lies above 0.046 mbar.
Moreover, new data on the $HO_2$ distributions were recently obtained from the MLS
measurements. Millán et al. (2015) performed the offline retrieval of daily zonal means of $HO_2$
profiles using averaged MLS radiances measured in 10° latitude bins. Averaged spectra have a
better signal to noise ratio, which removes many of the limitations of the MLS standard product for
$HO_2$. In particular, the upper boundary of the altitude region in which daytime data is suitable for
scientific use has reached 0.0032 mbar, and the "day-minus-night" correction is not needed at
altitudes above 1 mbar. Comparison with various experimental and model data has shown that the
offline retrieval reproduces the basic properties of the $HO_2$ distribution in the mesosphere relatively
well (at least qualitatively) (Millán et al. 2015).
The offline retrieval product, the alternative dataset of daytime $HO_2$, has recently become
publicly available at https://mls.jpl.nasa.gov. Figure 10 shows the monthly averaged zonal means
of offline retrieval data ($<HO_2^{MLS}{}_{offline}>$) and relative differences with retrieved and MLS standard
product data $(< HO_2^{MLS} > - < HO_2^{MLS}{}_{offline} >)/ < HO_2^{MLS}{}_{offline} >$ and
$(< HO_2^{ret} > - < HO_2^{MLS}{}_{offline} >)/ < HO_2^{MLS}{}_{offline} >$, correspondingly. Figure 10 represents the same time
periods as Figs. 6-8. It is worth noting that the distributions $< HO_2^{MLS}{}_{offline} >$ depicted in Fig. 10
represent significantly different amounts of data. The data sets for May and September include 31
and 27 days of measurements, respectively, whereas the January dataset encompasses only 4
days. The latter makes the graphs in the first row in Fig. 10 noisier than the others. One can see
that the results of the offline $HO_2$ retrieval show the same features as the results of our evaluation
technique in comparison to the standard MLS retrieval, i.e. the height of mesospheric $HO_2$
maximum is notably higher. We can conclude that the distributions of $< HO_2^{ret} >$ better match
$< HO_2^{MLS}{}_{offline} >$ than $< HO_2^{MLS} >$, although some quantitative discrepancy between $< HO_2^{ret} >$ and
$< HO_2^{MLS}{}_{offline} >$ also exists. Note that this may be due to systematic errors in the $HO_2^{MLS}$
distributions, which cannot be excluded within the framework of the introduced technique. For a
detailed qualitative and quantitative comparison of $< HO_2^{ret} >$ and $< HO_2^{MLS}{}_{offline} >$ one should
modify the method, so that a statistical evaluation of the $OH^{MLS}$ and $O_3^{MLS}$ standard products, and
the data of the offline $HO_2$ retrieval could be conducted within the framework of a single procedure
with no account for the $HO_2^{MLS}$ distributions. This modification is under way and will be presented
elsewhere.

The proposed method for statistical evaluation of mesospheric species measurements can

be readily generalized to other atmospheric photochemical systems that contain short-lived
components (see Introduction). It may also be modified for assessing hard to measure chemical
components, characteristics of atmospheric processes (like wind speed or turbulent diffusion rate),
or poorly known reaction rates.

**Acknowledgments**
This work was supported by the Russian Science Foundation (contract No. 15–17–10024 of June
04, 2015). The data used in this study is supported by the Institute of Applied Physics of the
Russian Academy of Sciences (Nizhny Novgorod, Russia). Inquiries about the distributions used in
this paper can be addressed to Mr. Belikovich (belikovich@ipfran.ru).

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

**Table 1**. List of reactions with corresponding reaction rates from Burkholder et al. (2015).


| 1 | $O(^1D)+O_2 \rightarrow O+O_2$ | 22 | $OH+O_3 \rightarrow O_2+HO_2$ | 43 | $NO_2+O_3 \rightarrow NO_3+O_2$ |
|---|---|---|---|---|---|
| 2 | $O(^1D)+N_2 \rightarrow O+N_2$ | 23 | $HO_2+O_3 \rightarrow OH+2O_2$ | 44 | $N+OH \rightarrow NO+H$ |
| 3 | $O(^1D)+O_3 \rightarrow O_2+2O$ | 24 | $H+OH+N_2 \rightarrow H_2O+N_2$ | 45 | $NO+HO_2 \rightarrow NO_2+OH$ |
| 4 | $O(^1D)+O_3 \rightarrow 2O_2$ | 25 | $OH+H_2 \rightarrow H_2O+H$ | 46 | $H +NO_2 \rightarrow OH+NO$ |
| 5 | $O(^1D)+N_2O \rightarrow 2NO$ | 26 | $OH+OH \rightarrow H_2O+O$ | 47 | $NO_3+NO \rightarrow 2NO_2$ |
| 6 | $O(^1D)+N_2O \rightarrow N_2+O_2$ | 27 | $OH+OH+M \rightarrow H_2O_2+M$ | 48 | $N+NO \rightarrow N_2+O$ |
| 7 | $O(^1D)+H_2O \rightarrow 2OH$ | 28 | $OH+HO_2 \rightarrow H_2O+O_2$ | 49 | $N+NO2 \rightarrow N2O+O$ |
| 8 | $O(^1D)+H_2 \rightarrow H+OH$ | 29 | $H_2O_2+OH \rightarrow H_2O+HO_2$ | 50 | $O_2+h\nu \rightarrow 2O$ |
| 9 | $O(^1D)+CH_4 \rightarrow CH_3+OH$ | 30 | $HO_2+HO_2 \rightarrow H_2O_2+O_2$ | 51 | $O_2+h\nu \rightarrow O+O(^1D)$ |
| 10 | $O(^1D)+CH_4 \rightarrow H_2+CH_2O$ | 31 | $HO_2+HO_2+M \rightarrow H_2O_2+O_2+M$ | 52 | $O_3+h\nu \rightarrow O_2+O$ |
| 11 | $O+O+M \rightarrow O_2+M$ | 32 | $CH_3+O \rightarrow CH_2O+H$ | 53 | $O_3+h\nu \rightarrow O_2+O(^1D)$ |
| 12 | $O+O_2+M \rightarrow O_3+M$ | 33 | $OH+CO \rightarrow H+CO_2$ | 54 | $N_2+h\nu \rightarrow 2N$ |
| 13 | $O+O_3 \rightarrow O_2 +O_2$ | 34 | $CH_4+OH \rightarrow CH_3+H_2O$ | 55 | $NO+h\nu \rightarrow N+O$ |
| 14 | $H+HO_2 \rightarrow 2OH$ | 35 | $CH_3+O_2+M \rightarrow CH_3O_2+M$ | 56 | $NO_2+h\nu \rightarrow NO+O$ |
| 15 | $H+HO_2 \rightarrow H_2O+O$ | 36 | $O_3+N \rightarrow NO+O_2$ | 57 | $N_2O+h\nu \rightarrow N_2+O(^1D)$ |
| 15 | $H+HO_2 \rightarrow H_2+O_2$ | 37 | $NO_3+O \rightarrow NO_2+O_2$ | 58 | $N_2O+h\nu \rightarrow N+NO$ |
| 17 | $OH+O \rightarrow H+O_2$ | 38 | $O+NO+M \rightarrow NO_2+M$ | 59 | $H_2O+h\nu \rightarrow H+OH$ |
| 18 | $HO_2+O \rightarrow OH+O_2$ | 39 | $NO_2+O \rightarrow NO+O_2$ | 60 | $CH_4+h\nu \rightarrow CH_2+H_2$ |
| 19 | $H_2O_2+O \rightarrow OH+HO_2$ | 40 | $NO_2+O+M \rightarrow NO_3+M$ | 61 | $H_2O_2+h\nu \rightarrow 2OH$ |
| 20 | $H+O_2+M \rightarrow HO_2+M$ | 41 | $N+O_2 \rightarrow NO+O$ | 62 | $NO_3+h\nu \rightarrow NO_2+O$ |
| 21 | $H+O_3 \rightarrow OH+O_2$ | 42 | $NO+O_3 \rightarrow NO_2+O_2$ | 63 | $CO_2+h\nu \rightarrow CO+O$ |




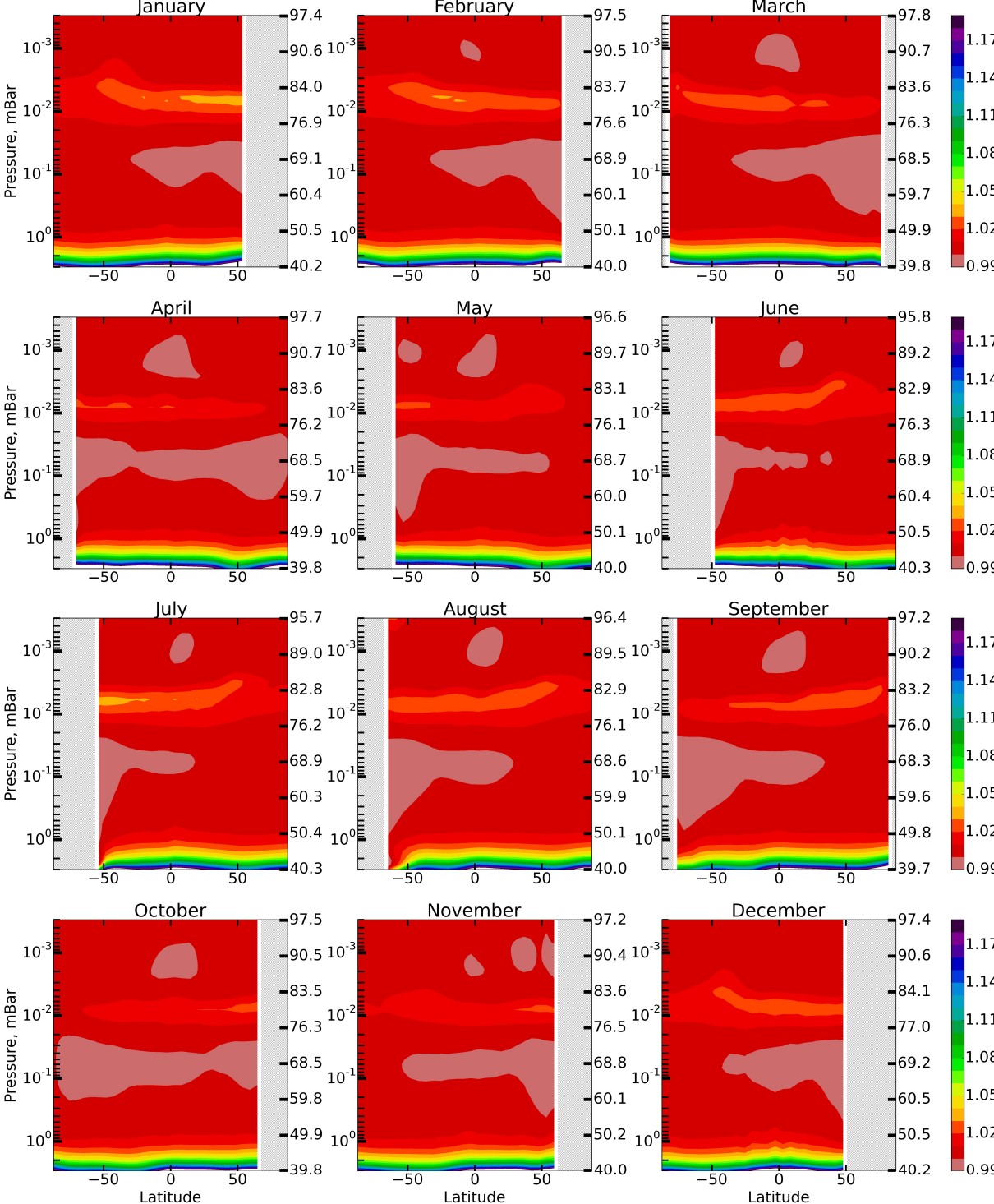


Figure 1. Daytime monthly averaged zonal mean $F$ distributions.


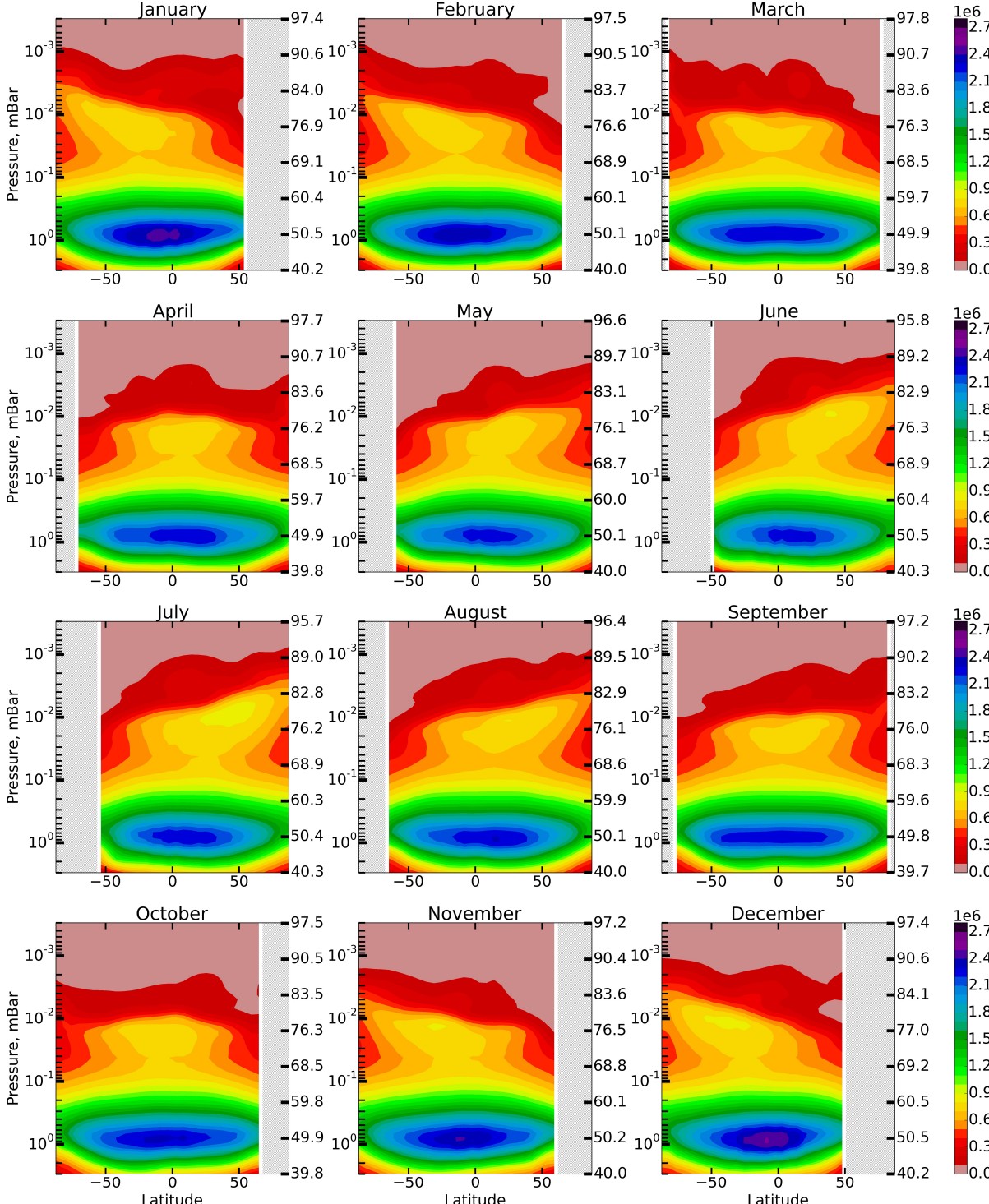

Figure 2. Daytime monthly averaged zonal mean $P_{OH}$ distributions (in cm$^{-3}$s$^{-1}$).





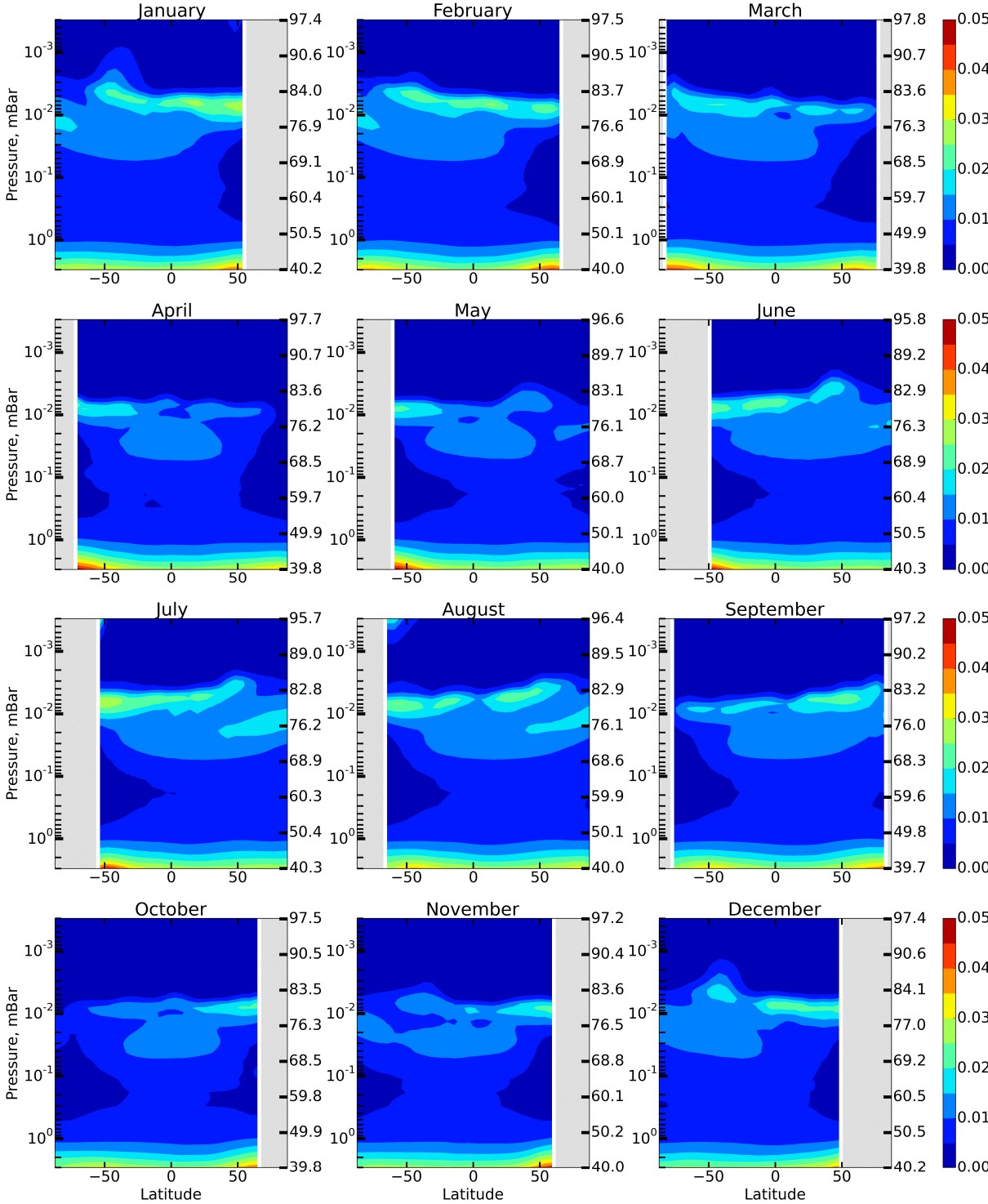



Figure 3. Daytime monthly averaged zonal mean $P_{OH}{}^{H_2O} / P_{OH}$ distributions.


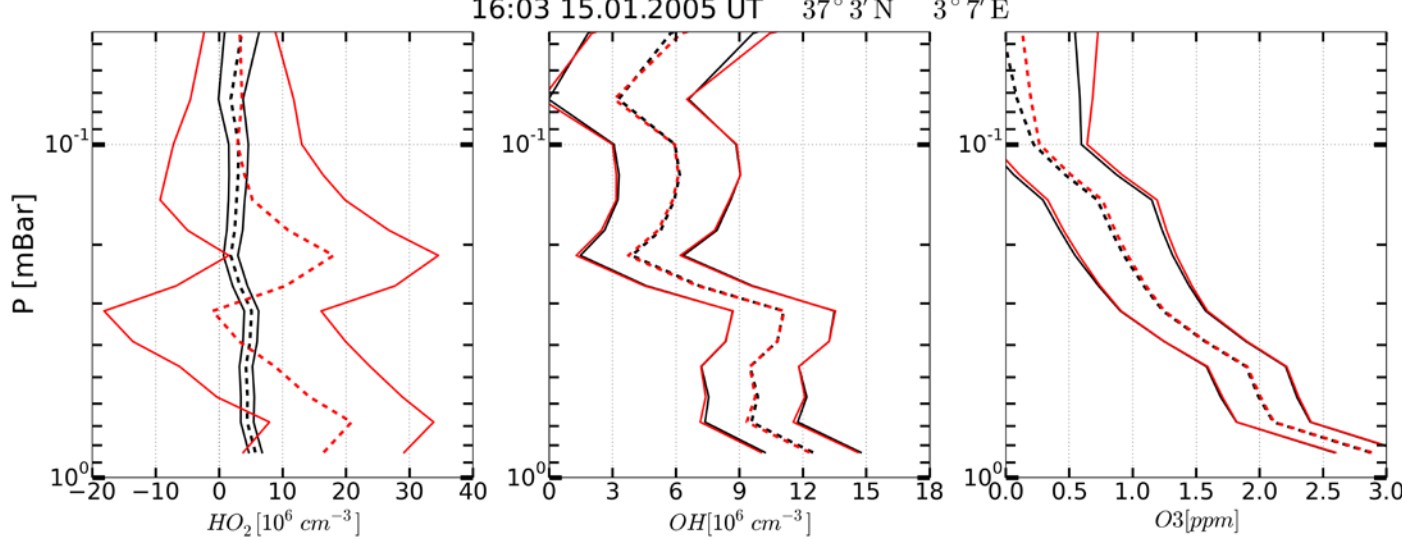



Figure 4. Example of OH, HO$_2$ and O$_3$ vertical profiles measured (red curves) on 15 January 2005
at 16.03 UT, 37$^0$3'N, 3$^0$7'E and corresponding retrieved profiles (black curves). Solid curves:
boundaries of the 65% confident intervals, dashed curves: medians.


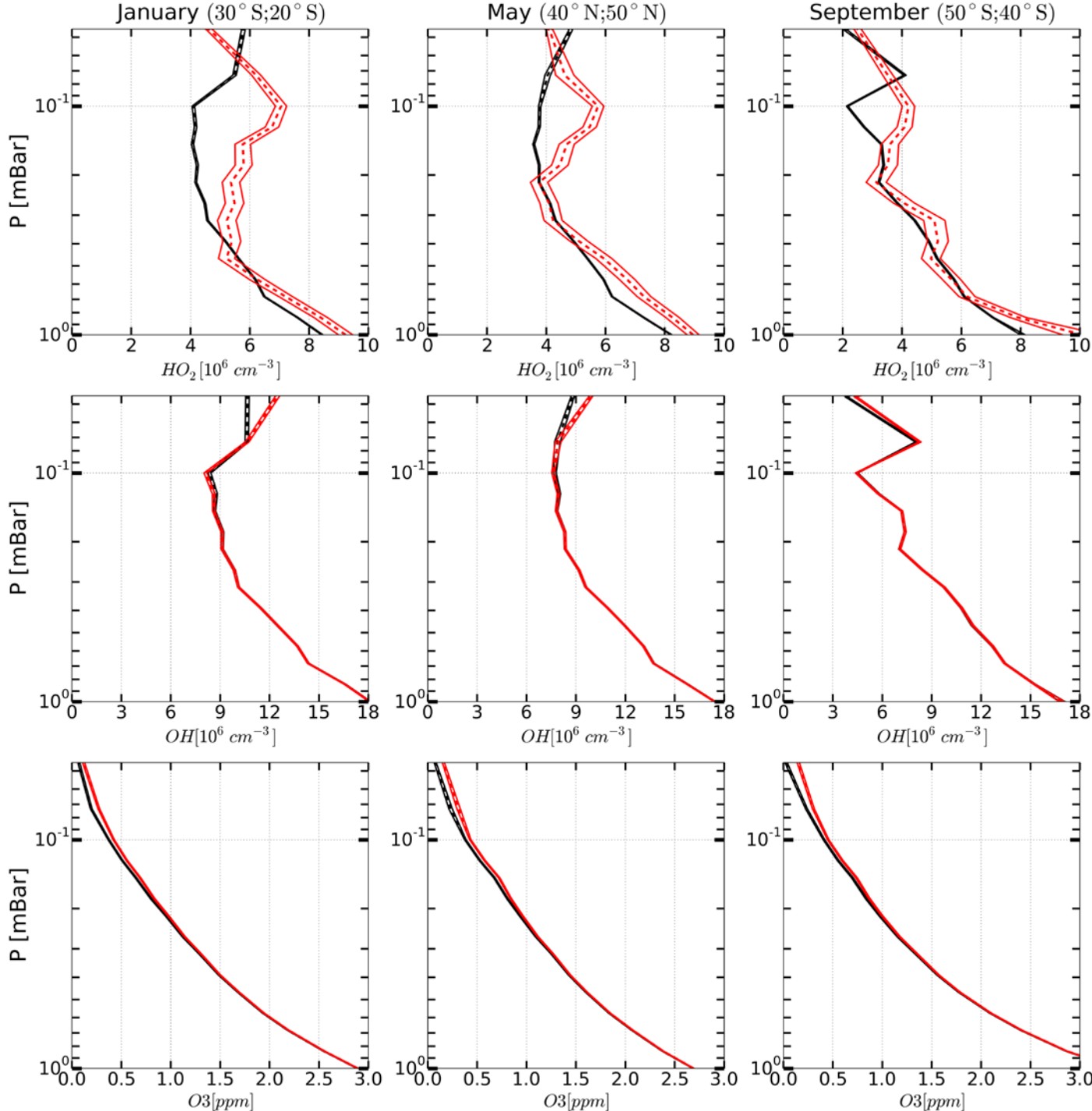



Figure 5. Examples of monthly averaged zonal mean vertical profiles of OH, HO₂ and O₃ measured
(red curves) in January, May and March 2005 and corresponding retrieved profiles (black curves).
Solid curves: boundaries of the 65% confident intervals, dashed curves: medians.

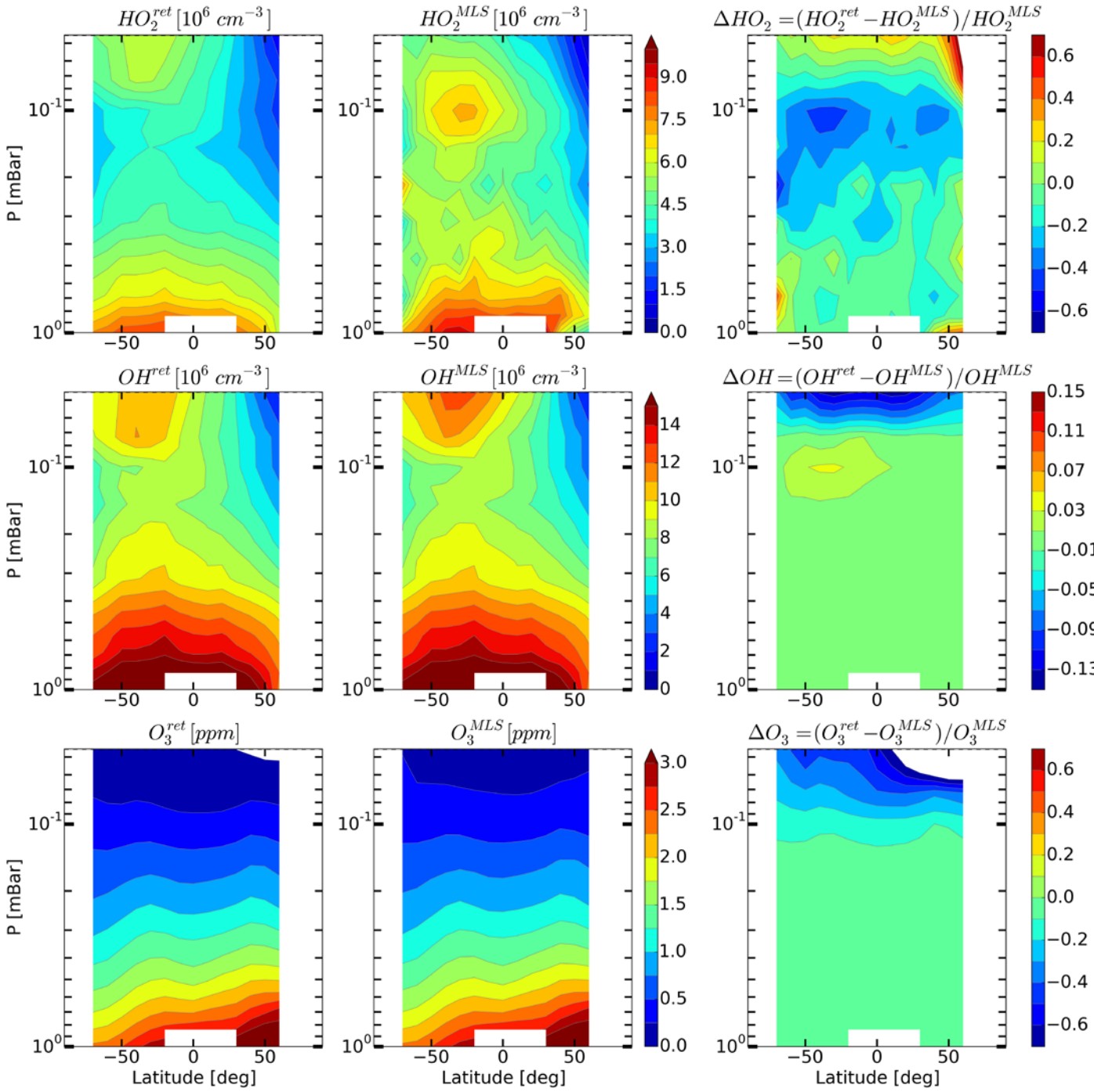


Figure 6. Daytime monthly averaged zonal mean retrieved (left column) and measured (middle column) distributions of HO$_2$, OH, and O$_3$ and their relative difference (right column) in January 2005.


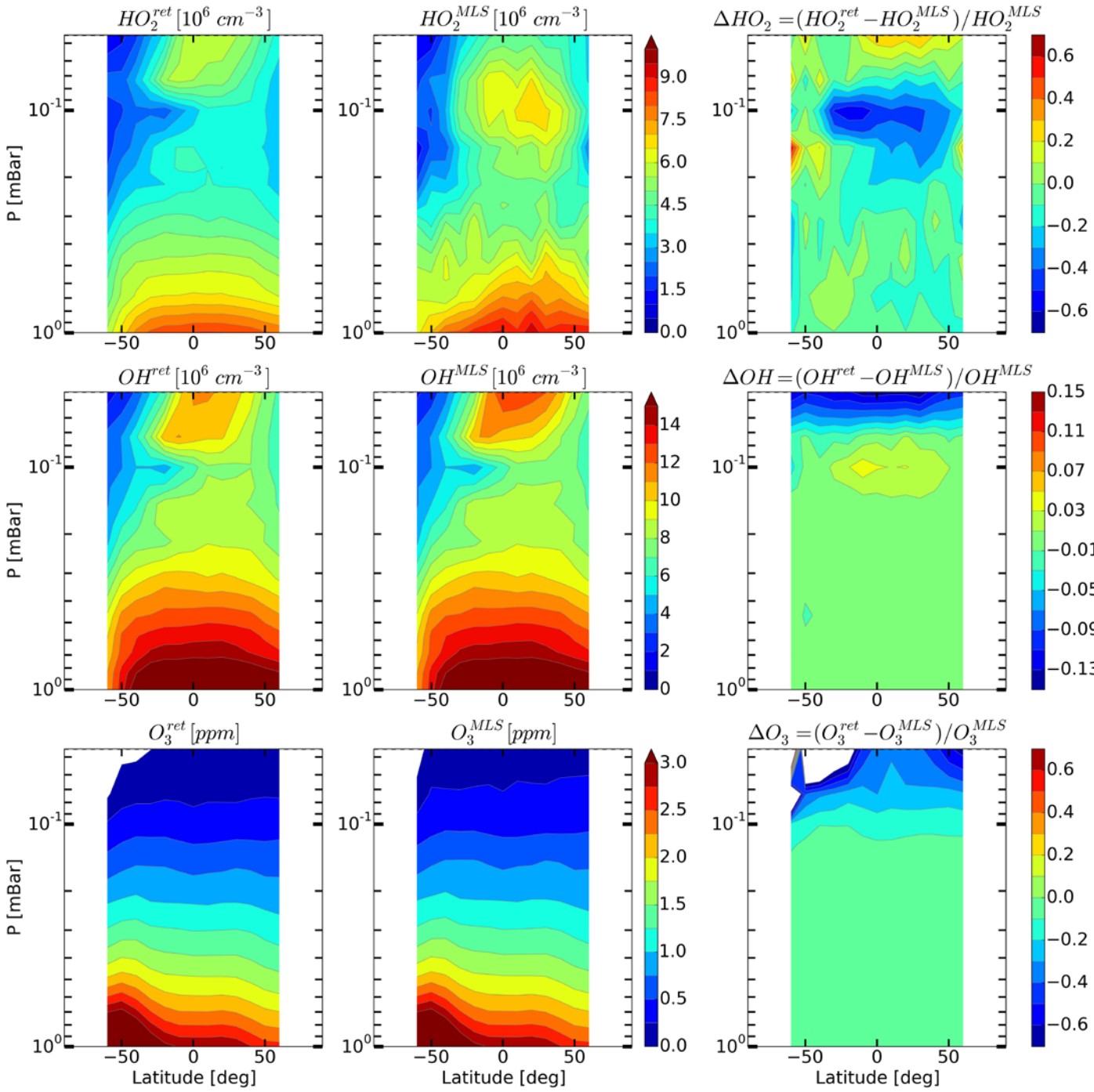

 Figure 7. Daytime monthly averaged zonal mean retrieved (left column) and measured (middle

 column) distributions of $HO_2$, OH, and $O_3$ and their relative difference (right column) for May 2005.


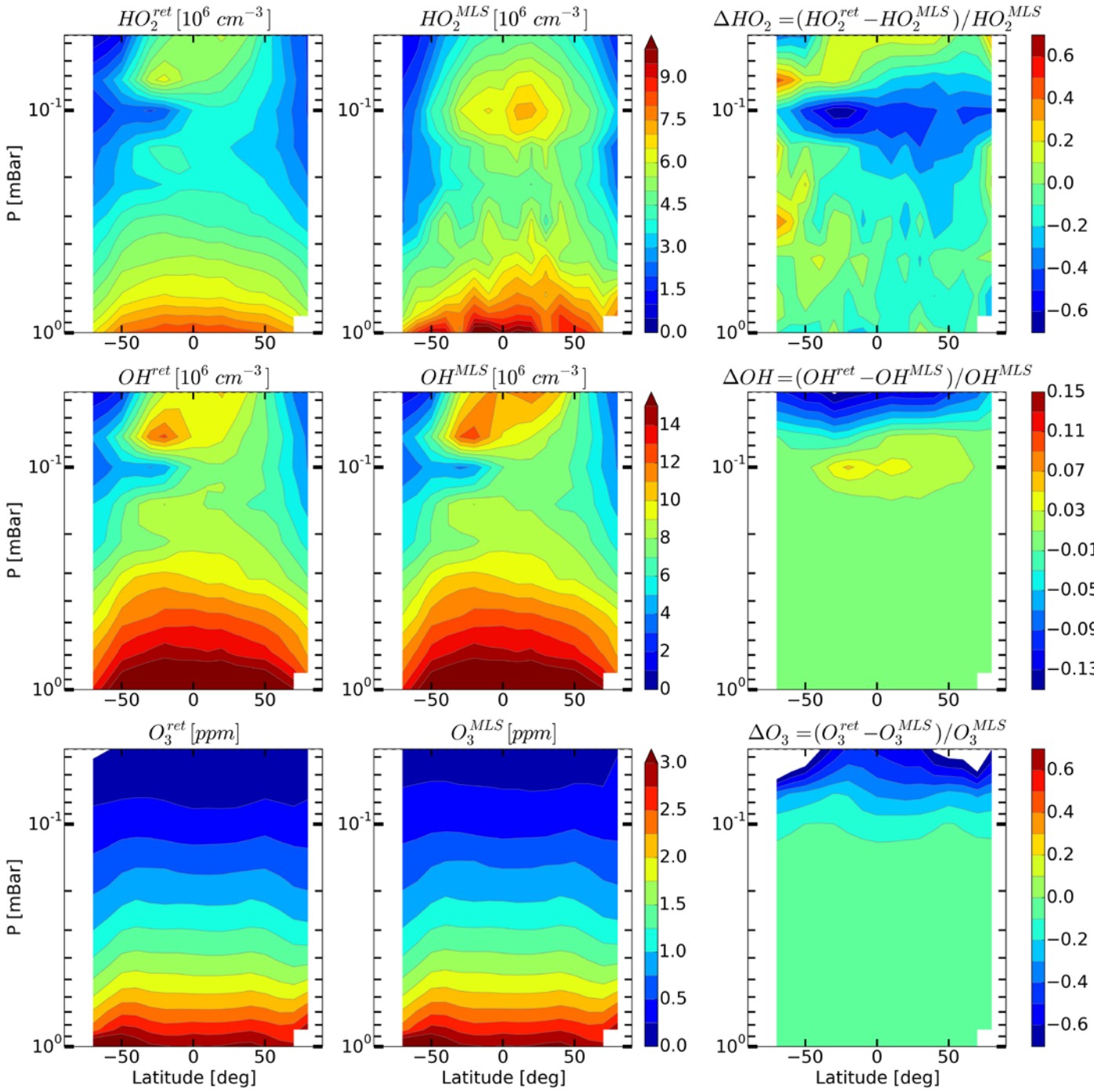

Figure 8. Daytime monthly averaged zonal mean retrieved (left column) and measured (middle
column) distributions of HO$_2$, OH, and O$_3$ and their relative difference (right column) for September
848 2005.


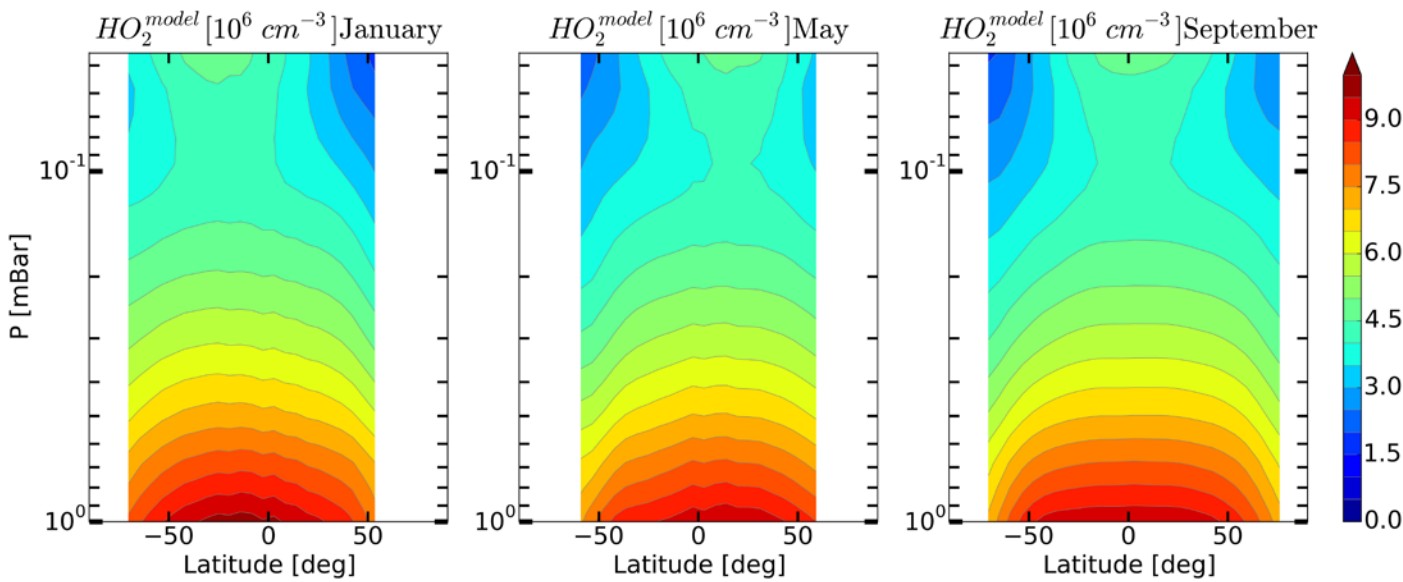


Figure 9. Daytime monthly averaged zonal mean model distributions of $HO_2$ for January, May, and
September.



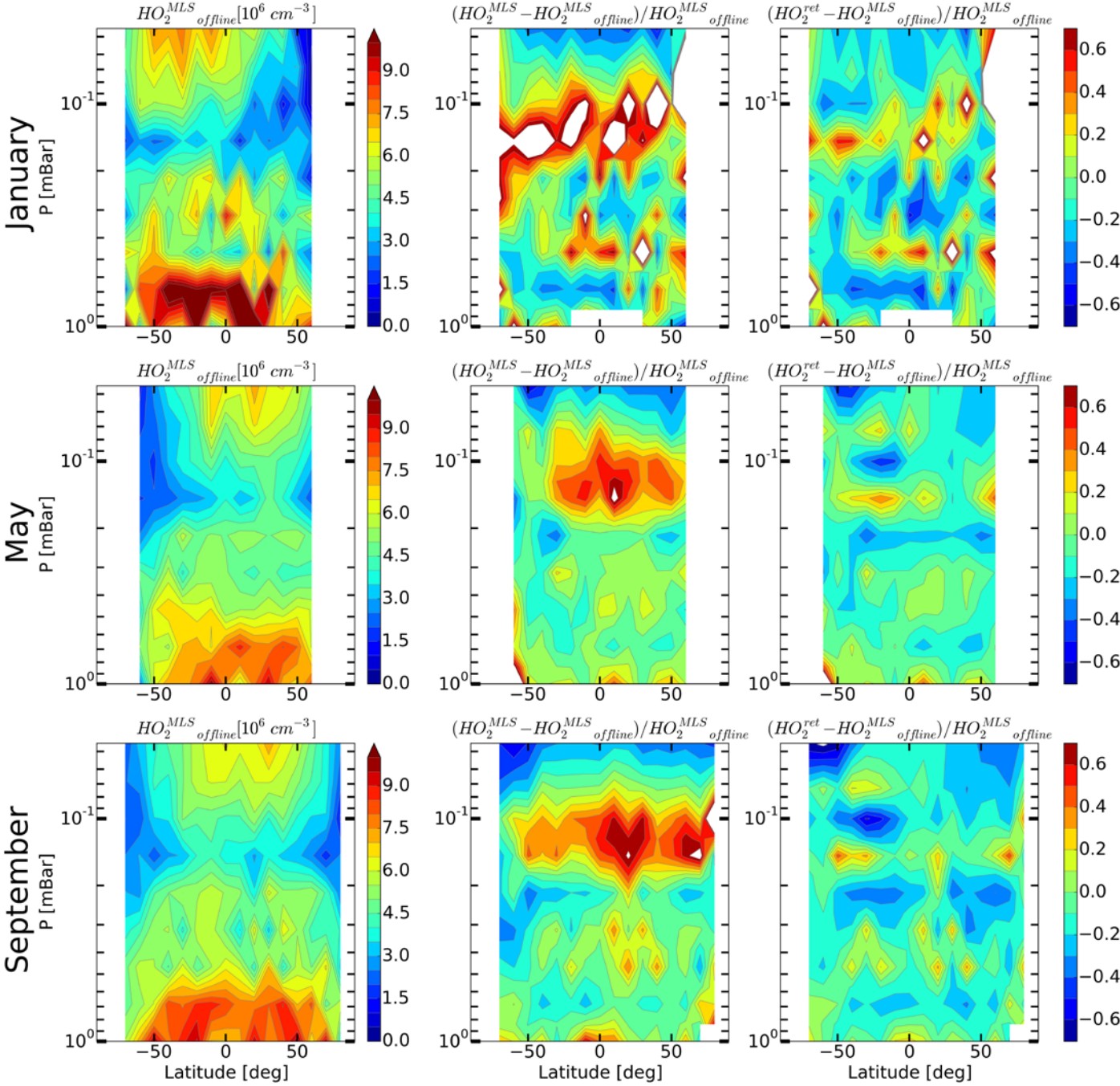



Figure 10. Daytime mean monthly averaged distributions of $HO_2$ retrieved by Millán et al. (2015)
and relative differences $(< HO_2^{MLS} > - < HO_2^{MLS}{}_{offline} >)/ < HO_2^{MLS}{}_{offline} >$ and
$(< HO_2^{ret} > - < HO_2^{MLS}{}_{offline} >)/ < HO_2^{MLS}{}_{offline} >$.