# Peer review of "Technical Note: Evaluation of simultaneous measurements of mesospheric OH, HO2, and O3"

_Atmospheric Chemistry and Physics, 2017_

## Referee Comment (RC1) · Anonymous Referee #1 · 15 Dec 2017

This technical note uses a statistical approach to the analysis of simultaneous measurements of OH, HO2, and O3 in the mesosphere from the Microwave Limb Sounder (MLS). A simplified algebraic equation is presented that relates these species under photochemical equilibrium, and this analytic formulation is compared to simulations from a chemical transport model. A Bayesian statistical approach is applied to the analytic formulation and constrained by the MLS measurements and their errors, in order to produce "retrieved" sets of trace gas profiles which are then compared with the original measurement profiles.

[Figure]

Overall, this methodology is interesting and appears to have some merit. However, there are a number of major issues that should be addressed in order to make this paper acceptable for publication.

Major Points:

1. There are a number of problems in the way the MLS data are presented and used. First, it is not clear that the day-night differences are being used for the "standard" HO2 product, as recommended in validation papers and in MLS data quality documents. This is critical since HO2 is a major focus for the paper. Second, there is no mention of the version of the data used (v3.3?) and whether all of the appropriate data screens were used. Third, all of the figures show data and profiles to 0.01 hPa, when in fact the useful range for HO2 is clearly stated at 0.046 hPa, and for O3 at 0.02 hPa. Data above these levels are not recommended for science purposes and should not be part of the analysis. This paper would clearly benefit from some interaction and discussion with members of the MLS team to ensure that the data are being applied correctly.

2. The connection between the CTM results and the analytic formulation is not sufficiently explored. Figure 1 is used to show that the difference between the CTM and equilibrium equation is a few percent or less, and that this justifies the use of the analytic formulation in the subsequent statistical evaluation. However, there is not enough information given about the CTM to judge whether or not this is really an independent validation. Presumably, the CTM uses the same set of reactions and the same equilibrium assumptions along with the "family" chemical species approach, so it is not surprising that they agree. It is also not clear whether the CTM also includes NOx chemistry, which might have an impact in the upper stratosphere and lower mesosphere. As a further example, there are known effects on HOx and NOx in the mesosphere from energetic particle precipitation events [e.g. many papers by Jackman et al and Verronen et al], and it would be important for the CTM to include these if it is intended to be used as a validation of the equilibrium expression. In fact, Figures 2 and 3 display data from January 2005, during a period with a documented SPE event [Jackman et

al., ACP, 2011]. The possible influence of SPE perturbations needs to be discussed and addressed.

If the CTM includes a comprehensive simulation of all known reactions and processes impacting HOx, and there is still agreement with the analytic approach, then the results are more credible. The paper should demonstrate that in fact the CTM provides a complete description of HOx photochemistry. Finally, there is a problem with converting pressure altitudes in the model to pressures by assuming a constant scale height of 7 km. This assumption can lead to offsets of 1 km or larger and is particularly critical when comparing vertical profiles with large gradients. For example, at 62 km the US std atmosphere has a pressure of 0.1671 mb. If a constant 7-km scale height is used in the expression in the paper, one would calculate p=0.1442 at 62 km. The analysis in this paper should use the temperatures in the model to compute convert altitude to pressure using hydrostatic equilibrium on a layer-by-layer basis.

3. There is not enough detail on the conclusion that the offline HO2 product better agrees with the statistical retrieval. First, there is no direct comparison between the offline product and the statistical retrieval, as there is for the standard product and the statistical retrieval (figs 4-6). There is only a presentation of the offline HO2 product, and the reader has to refer back to figs 4-6 and then estimate just how much better the agreement really is. The improvement needs to be quantified more directly. Also, there is no discussion as to why the agreement is better and this really gets down to the science. Does this paper intend to suggest that the offline HO2 product is superior to the standard product? If so, then the discussion/conclusion needs to be expanded to clearly justify why this is so. There also needs to be some explanation for why the offline product is different in the first place, without forcing to reader to go back and review the Milan et al [2015] paper.

Minor Points:

The grammar and wording is awkward or incorrect in a number of places, for example Abstract: "statistically correct approach" is not illuminating. One would not expect ACP to publish a "statistically incorrect approach". Suggest a "Bayesian statistical approach". "air concentration" is more generally "neutral density", "air temperature" is generally just "temperature". "We have performed \*a\* one-year simulation". Last sentence is unclear. Also "MLS primary data", what is primary data? Perhaps this should be MLS radiances.

In terms of the reaction sets, there appears to be a gaping omission of the HOx production reactions, H2O+hv and H2O+O(1D). These should be included for completeness; however, I suspect their terms may drop out when forming the ratio OH/HO2.

When deriving equation 14, which forms a major basis for the paper, I was not able to reproduce the authors' results and cannot comment on the validity of this result. It seems much more complicated than the text would suggest. I strongly recommend an appendix which details the step-by-step process at arriving at equation 14.

Figure 3 is unclear what ranges of latitudes are being shown.

Figs 4-6 contain horizontal dotted lines but these are not explained in the text or caption.

---

## Referee Comment (RC2) · Anonymous Referee #2 · 15 Dec 2017

I recommend the paper for publication in ACP after some minor corrections and changes. The concentrations of three constituents, OH, HO2, O3, have been measured by satellite and additionally calculated by a 3D global chemical transport model. The authors used MLS/Aura data in their paper inserting the measured concentrations into a derived formula. They apply a sophisticated statistic Bayesian validation technique. Likewise the concentrations of the same constituents have been calculated on the basis of the chemical transport model LIMA. The authors compared the results of both measurements and calculations and discuss the differences between them. In my

opinion a surprising result is that the derived expression does not explicitly depend on water vapor, but implicitly it depends on water vapor by the concentrations of the acting species.

In case of photochemical equilibrium, known temperature and pressure profile 5 non-linear algebraic equations determine the concentration of the chemical active species H, O, OH, OH2, and O3 in the MLT-region. Atomic oxygen and atomic hydrogen are the main active chemical constituents in this system. A further assumption in this study is that transports also can be neglected. The system depends then, in essential, on the variable concentration of water vapor. On condition that also the temperature-depending reaction rates and the dissociation rates are known, one could theoretically derive, if the concentration of only one constituent was measured, the concentration of water vapor or any other constituuent.

Photochemical equilibrium occurs in the MLT-region only during daytime. If more than one constituent were measured one has the opportunity to check model calculations or to inspect the agreement to observed data. On the one hand the authors use for analysis in their paper satellite measured concentrations of OH, HO2 and O3. Neglecting the small term JH2O[H2O] in the equation for OH (equation 10) both main chemical active constituents, O and H, can stepwise be eliminated from the system (first O in equation 13), and an expression of the structure F(OH, HO2, O3)=1 can be derived only depending on the measured constituents. This equation depends in nonlinear way on OH, HO2 and O3.

On the one hand the authors use model results of 3D-calculations and on the other hand they employ measured concentrations of these species. Deviations from unity are a hint to non-equilibrium conditions or to other reasons such as incorrect reaction rates, erroneous model calculations or errors in the retrieved data. The fundamental idea was to derive such expression not depending on water vapor. The interpretation of deviations from unity is certainly complicated. What means for instance -1% deviation or F(OH, HO2, O3)=0.99? It indicates that, generally speaking, the agreement is quite

good. But in case of stronger deviations it is difficult to say, what is the reason for this discrepancy? Minor comments:

The paper has a very voluminous introduction of about 39% related to the entire paper. It has the character of a review paper. I will it not criticize that, but I will it only mention here.

Line 165: 150 km is already the middle thermosphere.

Line 174: entered, maybe better mentioned

Line 178: H=7 km is an approximated mean scale height. (1 km scale height corresponds to about 33 K or 7 km to 231 K mean temperature)

Line 194: reaction rate constants; according to? Quotation?

Line 198: The net production term of hydrogen radicals is in essential $JH_2O[H_2O]$. Why do you neglect this term in (10)? Too small compared with the other terms in equation (10) and consequently the approach do not depend on water vapor?

Line 202: ...of ozone ()?

Line 204: The aim is to eliminate O and H and to derive an expression only depending on OH, $HO_2$, $O_3$.

Line 211: $\alpha=(\ldots)$ could be equation (14.2) and (14) then (14.1) or (15) and the following equations (x+1).

There is a large step from equation (10) – (13) to equation (14). Could you give some intermediate steps?

Line 218: k2 decreases strongly below the lower mesosphere and stratopause. Ozone is no longer in photochemical equilibrium there.

Line 234: ...certain altitude z...

Line: 239: Factor $\sigma j\sqrt{2}\,\pi$?

Line 296: . . .fall into one. . .

Line 304-318: In the lower thermosphere the system is not in chemical equilibrium. Transports play a significant role (see also Grygalashvyly et al. 2012).

Section 6: The characteristic time of atomic oxygen is about $\tau O = (k1 O2 M)^{-1}$. At 90 km is $O2 = 1.47 \times 10^{13}$ cm-3, $M = 7 \times 10^{13}$ cm $^{-3}$, and $k1 \approx 10^{-33}$ cm6s-1 depending on temperature. The characteristic time has then an order of $10^6$ s. About one order smaller is the characteristic time of H, but still large. Both the production and the loss term of HO2 depend on H and O being not in photochemical equilibrium in the lower thermosphere. Therefore a discrepancy relating to HO2 one should expect.
* * *

---

## Author Comment (AC1) · 31 Mar 2018

Response to the comments on the paper by Referee #1

**Major Points:**

1. *There are a number of problems in the way the MLS data are presented and used. First, it is not clear that the day-night differences are being used for the "standard" HO2 product, as recommended in validation papers and in MLS data quality documents. This is critical since HO2 is a major focus for the paper. Second, there is no mention of the version of the data used (v3.3?) and whether all of the appropriate data screens were used. Third, all of the figures show data and profiles to 0.01 hPa, when in fact the useful range for HO2 is clearly stated at 0.046 hPa, and for O3 at 0.02 hPa. Data above these levels are not recommended for science purposes and should not be part of the analysis. This paper would clearly benefit from some interaction and discussion with members of the MLS team to ensure that the data are being applied correctly.*

We agree with this comment. In the revised manuscript the following corrections are made (see lines 322-333):

(1) We have used the day-minus-night differences for the $HO_2$ product. Accordingly, the results presented in Figs. 2–6 have been completely recalculated.

(2) The information on the MLS data version in use, i.e. v4.2, the latest one, have been added. All the data screens were applied when dealing with the MLS standard product.

(3) We have limited our analysis to the 1–0.046 mbar pressure interval where all data are suitable for scientific use, as prescribed by (Wang et al., 2015; Livesey et. al., 2017).

2. *The connection between the CTM results and the analytic formulation is not sufficiently explored. Figure 1 is used to show that the difference between the CTM and equilibrium equation is a few percent or less, and that this justifies the use of the analytic formulation in the subsequent statistical evaluation. However, there is not enough information given about the CTM to judge whether or not this is really an independent validation. Presumably, the CTM uses the same set of reactions and the same equilibrium assumptions along with the "family" chemical species approach, so it is not surprising that they agree. It is also not clear whether the CTM also includes NOx chemistry, which might have an impact in the upper stratosphere and lower mesosphere.*

We agree that the original version of the manuscript contained little information about the 3D chemical transport model (CTM). Indeed, our model uses a family concept by Shimazaki (Shimazaki, 1985). It is used to calculate the evolution of the components of

HOx and NOx families, while the Ox family is calculated via regular implicit Euler method. Moreover, it should be emphasized that the Shimazaki scheme utilizes an implicit Euler scheme too and does not use the steady state approximation for short-lived components. Thus, in calculating the evolution of OH, $HO_2$, and $O_3$ within the framework of the CTM, we *do not use* the photochemical equilibrium condition.

In the revised manuscript the following changes have been made:

(1) A list of reactions, accounted for by the CTM, is added (see Table 1). It can be seen first of all, that the CTM includes comprehensive NOx chemistry. Secondly, the complete set of reactions (63 in total) is much bigger than the one we consider to describe the daytime balance of OH, $HO_2$, and $O_3$ concentrations. In the original manuscript there were 9 of them — now there are 8, the reaction $OH + HO_2 \rightarrow H_2O + O_2$ was removed. Our numerical analysis showed that its contribution to the analytic expression $F(OH, HO_2, O_3) = 1$ is less than 1%.

(2) The description of the chemical transport model, including its dynamics and the integration methods used, is substantially expanded (see lines 167-195). In particular, it is stated that «The evolution of the components of $HO_x$ (H, OH, $HO_2$, $H_2O_2$) and $NO_x$ (N, NO, $NO_2$, $NO_3$) families is calculated using the chemical family concept proposed by Shimazaki (Shimazaki, 1985). This is done because of the presence of short-lived components among these families, with lifetimes much shorter than those of the families themselves, which imposes significant restrictions on the value of the CTM's integration step. For example, the daytime lifetimes of OH and $HO_2$ above 70 km are about 1 s or less, while the lifetime of the $HO_x$ family is about $10^4$ s or more. Therefore, when calculating these components individually it is necessary to set the CTM's integration step to be much less than 1 s. In our work, the Shimazaki technique is applied for calculating the evolution of each component of the $HO_x$ and $NO_x$ families. We emphasize that this technique does not explicitly use the steady-state approximation for the components, instead it utilizes the approach based on an implicit Euler scheme (see Shimazaki, 1985). This allows increasing the integration step of CTM significantly without loss of accuracy of calculating the short-lived components. In our work the integration time is chosen to be 9 s.»

*As a further example, there are known effects on HOx and NOx in the mesosphere from energetic particle precipitation events [e.g. many papers by Jackman et al and Verronen et al], and it would be important for the CTM to include these if it is intended to be used as a validation of the equilibrium expression. In fact, Figures 2 and 3 display data from January*

*2005, during a period with a documented SPE event [Jackman etal., ACP, 2011]. The possible influence of SPE perturbations needs to be discussed and addressed.*

Done. We carried out a brief analysis of the possible effect of solar proton events (SPE), which were not implemented in the CTM, on the mesospheric photochemistry in daytime. We considered the most prominent impact of SPE in the context of the problem: the impact on the chemical balance of OH. Comparing the additional OH source due to SPE with the main source of this component (via the reactions $HO_2 + O \to OH + O_2$, $H + O_3 \to OH + O_2$ and $H + HO_2 \to 2OH$) we showed that the influence of SPE on the daytime balance of OH is insignificant.

We made two additions to the revised manuscript:

(1) a new figure (see Fig. 2);

(2) the following text (see lines 251-256, 263-275):

«Note also that Eq. (1) and Eq. (6) take into account only the main daytime source of OH ($P_{OH}$) specified by reactions R18, R14, and R21:

$$P_{OH} = k_{18} \cdot HO_2 \cdot O + 2k_{14} \cdot HO_2 \cdot H + k_{21} \cdot O_3 \cdot H$$

These reactions run "inside" the HOx (H, OH, $HO_2$, $H_2O_2$) family and do not perturb its total concentration. The height–latitude cross-sections of $<P_{OH}>$ for each month are presented in Fig. 2.

….

Another source of OH is sporadically activated during charged particle precipitation events and exists for a relatively short time (several days). Solar proton events (SPE) perturb the ionic composition in the mesosphere and the upper stratosphere considerably and trigger a whole cascade of reactions involving ions, neutral components and their clusters (e.g., $O_2^+ \cdot H_2O$). This leads to an additional (to reactions R59 and R7) conversion of $H_2O$ molecules into OH and H (Solomon et al., 1981). The maximum of the OH production rate ($P_{OH}^{SPE}$) induced by SPE is located in the polar latitudes in the region of 60–80 km and, as a rule, does not exceed $2 \cdot 10^3$ cm$^{-3}$ s$^{-1}$ (Jackman et al., 2011, 2014). It can be seen from Fig. 2 that at these latitudes and altitudes the $P_{OH}^{SPE} / P_{OH}$ ratio does not exceed 1-2%, even for the maximum values of $P_{OH}^{SPE}$. This means that the impact of $P_{OH}^{SPE}$ on Eq. (6) is of the same order of smallness as in the case of reactions R59 and R7, hence, it may be neglected. A similar conclusion can be made for other reactions from Table 1, not accounted for by Eq. (6), including the ones involving NO$_x$ in both quiet and perturbed conditions in the mesosphere.»

(3) We found the mistake in Fig. 1 below 50 km caused by the use of improper computer number format (float32 instead of float64). So Fig.1 was recalculated and redrawn. The value of $<F>$ below 50 km increases.

*If the CTM includes a comprehensive simulation of all known reactions and processes impacting HOx, and there is still agreement with the analytic approach, then the results are more credible. The paper should demonstrate that in fact the CTM provides a complete description of HOx photochemistry.*

Done. To sum up, the following corrections have been made:

(1) A list of reactions is added, accounted for by the CTM. It can be seen that the CTM includes complete chemistry of the mesosphere.

(2) The description of the chemical transport model is substantially expanded in the part relating to dynamics and the methods of integration applied.

(3) Analysis of the possible effect of solar proton events (SPE), which are not implemented in our model, on the photochemistry in the daytime mesosphere is carried out.

*Finally, there is a problem with converting pressure altitudes in the model to pressures by assuming a constant scale height of 7 km. This assumption can lead to offsets of 1 km or larger and is particularly critical when comparing vertical profiles with large gradients. For example, at 62 km the US std atmosphere has a pressure of 0.1671 mb. If a constant 7-km scale height is used in the expression in the paper, one would calculate p=0.1442 at 62 km. The analysis in this paper should use the temperatures in the model to compute convert altitude to pressure using hydrostatic equilibrium on a layer-by-layer basis.*

Done. In the revised manuscript, the pseudo-height scale was replaced by pressure levels (see lines 199-202 and Figs 1-3). In addition, we indicated the approximate heights in km which were calculated for a given month utilizing averaged temperature profiles of the model and hydrostatic equilibrium.

*3. There is not enough detail on the conclusion that the offline HO2 product better agrees with the statistical retrieval. First, there is no direct comparison between the offline product and the statistical retrieval, as there is for the standard product and the statistical retrieval (figs 4-6). There is only a presentation of the offline HO2 product, and the reader has to refer back to figs 4-6 and then estimate just how much better the agreement really is. The improvement needs to be quantified more directly. Also, there is no discussion as to why the agreement is better and this really gets down to the science. Does this paper intend to*

*suggest that the offline HO2 product is superior to the standard product? If so, then the discussion/conclusion needs to be expanded to clearly justify why this is so. There also needs to be some explanation for why the offline product is different in the first place, without forcing to reader to go back and review the Milan et al [2015] paper.*

Done. First of all, the description of the offline $HO_2$ retrieval is expanded and its advantages over the standard MLS product are pointed out (see lines 395-403):

«Moreover, new data on the $HO_2$ distributions were recently obtained from the MLS measurements. Millán et al. (2015) performed the offline retrieval of daily zonal means of $HO_2$ profiles using averaged MLS radiances measured in 10° latitude bins. Averaged spectra have a better signal to noise ratio, which removes many of the limitations of the MLS standard product for $HO_2$. In particular, the upper boundary of the altitude region in which daytime data is suitable for scientific use has reached 0.0032 mbar, and the "day-minus-night" correction is not needed at altitudes above 1 mbar. Comparison with various experimental and model data has shown that the offline retrieval reproduces the basic properties of the $HO_2$ distribution in the mesosphere relatively well (at least qualitatively) (Millán et al. 2015).»

Second, we made a direct comparison (see Fig. 10) of the offline MLS product with the results of our statistical retrieval and the standard product of MLS. One can see from Fig. 10 that our results better match the offline product than the standard one. The most noticeable difference is in the location of the mesospheric maximum of the $HO_2$ concentration. According to the standard product it is close to 0.1 mbar, while the retrieved data and the offline product demonstrate the altitudes above 0.046 mbar. This is also confirmed by the $HO_2$ distributions calculated using our 3D chemical transport model (see lines 391-394 and Fig. 9), In the revision we highlighted (see lines) that the higher location of that maximum in the results of the of $HO_2$ statistical retrieval is due to the influence of MLS data on OH, which have the mesospheric maximum (see Figs. 6–8) also well above 0.1 mbar.

**Minor Points:**

*The grammar and wording is awkward or incorrect in a number of places, for example Abstract: "statistically correct approach" is not illuminating. One would not expect ACP to publish a "statistically incorrect approach". Suggest a "Bayesian statistical approach".*

Corrected. "statistically correct approach" was replaced by "statistical approach" everywhere in text. See lines 14, 135.

*"air concentration" is more generally "neutral density",*

Corrected. "air concentration" was replaced by "neutral density" everywhere in text. See lines 19, 141.

*"air temperature" is generally just "temperature".*

Corrected. "air temperature" was replaced by "temperature" everywhere in text. See lines 19, 141, 323.

*"We have performed \*a\* one-year simulation".*

Corrected. See line 20.

*Last sentence is unclear. Also "MLS primary data", what is primary data? Perhaps this should be MLS radiances.*

Corrected. See lines 29-30.

*In terms of the reaction sets, there appears to be a gaping omission of the HOx production reactions, H2O+hv and H2O+O(1D). These should be included for completeness; however, I suspect their terms may drop out when forming the ratio OH/HO2.*

Done. We have directly compared the source of OH ($P_{OH}{}^{H_2O}$) due to the reactions $H_2O + hv \rightarrow H + OH$ and $O(^1D) + H_2O \rightarrow 2OH$ with the main source of that component, $P_{OH}$, via the reactions $HO_2 + O \rightarrow OH + O_2$, $H + O_3 \rightarrow OH + O_2$ and $H + HO_2 \rightarrow 2OH$. The ratio $P_{OH}{}^{H_2O} / P_{OH}$ does not exceed 3-4%.

Two corrections are made in the revised manuscript:

(1) new figures are added (Fig. 2-3);

(2) the following text is inserted (lines 251-262):

«Note also that Eq. (1) and Eq. (6) take into account only the main daytime source of OH ($P_{OH}$) specified by reactions R18, R14, and R21:

$$P_{OH} = k_{18} \cdot HO_2 \cdot O + 2k_{14} \cdot HO_2 \cdot H + k_{21} \cdot O_3 \cdot H$$

These reactions run "inside" the HOx (H, OH, HO₂, H₂O₂) family and do not perturb its total concentration. The height–latitude cross-sections of $< P_{OH} >$ for each month are presented in Fig. 2.

The next important daytime source of OH is specified by reactions R59 and R7 involving $H_2O$, the main source for the $HO_x$ family:

$$P_{OH}{}^{H_2O} = (k_{59} + 2 \cdot k_7 \cdot O(^1D)) \cdot H_2O$$

Figure 3 shows height–latitude cross-sections of $< P_{OH}{}^{H_2O} / P_{OH} >$ for each month. Comparing Fig. 1 and Fig. 3, we conclude that the previously indicated 3–4 % deviation of $<F>$ from 1 in the region between 76 km and 86 km is largely due to the neglect of these reactions. »

*When deriving equation 14, which forms a major basis for the paper, I was not able to reproduce the authors' results and cannot comment on the validity of this result. It seems much more complicated than the text would suggest. I strongly recommend an appendix which details the step-by-step process at arriving at equation 14.*

The whole section (Sec.2) was rewritten at the price of an insignificant increase in volume. Including, all the steps needed to derive the equation $F(OH, HO_2, O_3) = 1$ are presented. We believe that in this form this section harmoniously fits into the main canvas.

*Figure 3 is unclear what ranges of latitudes are being shown.*

Corrected. See Fig. 5.

*Figs 4-6 contain horizontal dotted lines but these are not explained in the text or caption.*

This horizontal dotted lines marked the upper limit of air pressure (0.046 mbar) where $HO_2$ data are suitable for scientific use. In the revised manuscript we restricted the pressure range by 0.046 mbar.

---

## Author Comment (AC2) · 31 Mar 2018

Response to the comments on the paper by Referee #2

**Main text:**

*On the one hand the authors use model results of 3D-calculations and on the other hand they employ measured concentrations of these species. Deviations from unity are a hint to non-equilibrium conditions or to other reasons such as incorrect reaction rates, erroneous model calculations or errors in the retrieved data. The fundamental idea was to derive such expression not depending on water vapor. The interpretation of deviations from unity is certainly complicated. What means for instance -1% deviation or F(OH, HO$_2$, O$_3$)=0.99? It indicates that, generally speaking, the agreement is quite good. But in case of stronger deviations it is difficult to say, what is the reason for this discrepancy?*

In the revised manuscript (see Fig. 3 and lines 251-262), we have directly compared the source of OH ($P_{OH}^{H_2O}$) due to the reactions H$_2$O+$hv$ → H+OH и O($^1$D)+H$_2$O → 2OH with the main source of that component, $P_{OH}$, via the reactions HO$_2$ + O → OH + O$_2$, H + O$_3$ → OH + O$_2$ and H + HO$_2$ → 2OH. It can be seen from Fig. 1 and Fig. 3 that the indicated 3–4 % deviation of $<F>$ from 1 in the region between 76 km and 86 km is largely due to the neglect of the reactions H$_2$O+$hv$ → H+OH и O($^1$D)+H$_2$O → 2OH.

Also, one should note the following. We found the mistake in Fig. 1 below 50 km caused by the use of improper computer number format (float32 instead of float64). The value of $<F>$ below 50 km increases.

**Minor comments:**

*The paper has a very voluminous introduction of about 39% related to the entire paper. It has the character of a review paper. I will it not criticize that, but I will it only mention here.*

Indeed, the Introduction contains elements of the review. Thus, we wanted to show that the proposed method of evaluation of simultaneous measurements of mesospheric components has a wide range of possible applications for other areas of the atmosphere.

*Line 165: 150 km is already the middle thermosphere.*

Corrected. See line 166.

Corrected. See line 197.

Corrected (see lines 197-202 and Figs 1-3). In the revised manuscript, the pseudo-height scale was replaced by pressure levels. In addition, we indicated the approximate heights in km calculated from the pressure profiles for a given month utilizing averaged temperature profiles of the model and hydrostatic equilibrium.

Corrected. See line 227 and Table 1.

In the revised manuscript, we have directly compared the source of OH ($P_{OH}^{H_2O}$) due to the reactions $H_2O+h\nu \rightarrow H+OH$ and $O(^1D)+H_2O \rightarrow 2OH$ with the main source of that component, $P_{OH}$, via the reactions $HO_2 + O \rightarrow OH + O_2$, $H + O_3 \rightarrow OH + O_2$ and $H + HO_2 \rightarrow 2OH$. The ratio $P_{OH}^{H_2O} / P_{OH}$ does not exceed 3-4%.

Two corrections are made in the revised manuscript:
(1) a new figure is added (Fig. 2-3);
(2) the following text is inserted (lines 251-262):
«Note also that Eq. (1) and Eq. (6) take into account only the main daytime source of OH ($P_{OH}$) specified by reactions R18, R14, and R21:

$$P_{OH}=k_{18} \cdot HO_2 \cdot O + 2k_{14} \cdot HO_2 \cdot H + k_{21} \cdot O_3 \cdot H$$

These reactions run "inside" the HOx (H, OH, HO$_2$, H$_2$O$_2$) family and do not perturb its total concentration. The height–latitude cross-sections of $< P_{OH} >$ for each month are presented in Fig. 2.
The next important daytime source of OH is specified by reactions R59 and R7 involving H$_2$O, the main source for the HO$_x$ family:

$$P_{OH}{}^{H_2O}=(k_{59} + 2 \cdot k_7 \cdot O(^1D)) \cdot H_2O$$

Figure 3 shows height–latitude cross-sections of $<P_{OH}{}^{H_2O} / P_{OH}>$ for each month. Comparing Fig. 1 and Fig. 3, we conclude that the previously indicated 3–4 % deviation of $<F>$ from 1 in the region between 76 km and 86 km is largely due to the neglect of these reactions.»

*Line 202: ...of ozone ()?*
Corrected. See line 230.

*Line 204: The aim is to eliminate O and H and to derive an expression only depending on OH, HO2, O3.*
Done. We added new sentence (see line 228):
«We eliminate O and H from Eqs. (1)-(3) and derive an expression depending only on OH, HO$_2$, O$_3$.»

*Line 211: α=(...) could be equation (14.2) and (14) then (14.1) or (15) and the following equations (x+1). There is a large step from equation (10) – (13) to equation (14). Could you give some intermediate steps?*
Done. The Sec.2 was rewritten. All the steps needed to derive the equation $F(OH,HO_2,O_3)=1$ were presented. It should be noted that the reaction OH + HO$_2$ $\rightarrow$ H$_2$O + O$_2$ was removed from the analysis. Our numerical calculations showed that its contribution to the analytic expression $F(OH,HO_2,O_3)=1$ is less than 1%.

*Line 218: k2 decreases strongly below the lower mesosphere and stratopause. Ozone is no longer in photochemical equilibrium there.*
We cannot fully agree with the comment. In particular, daytime lifetime of O$_3$ in the altitude range of 30-50 km varies in the range 100-1000 s (Brasseur and Solomon, 2005). So, O$_3$ can stay in photochemical equilibrium depending on height and duration of daylight.
The following sentences were inserted in the revised manuscript (see lines 247-250):
„Note that these components remain short-lived below 50 km (with the lifetimes of about $10^2$-$10^3$ s (Brasseur and Solomon, 2005)) depending on height and duration of daylight. However, for quantitative description of their daytime equilibrium it is

necessary to include additional reactions involving, in particular, the components of the NO$_x$ family."

*Line 234:...certain altitude z...*
Corrected. See line 290.

*Line: 239: Factor σj√2π?*
Corrected. See line 295.

*Line 296:...fall into one...*
Corrected. See line 360.

*Line 304-318: In the lower thermosphere the system is not in chemical equilibrium. Transports play a significant role (see also Grygalashvyly et al. 2012).*
*Section 6: The characteristic time of atomic oxygen is about $\tau O=(k1*O_2*M)^{-1}$ . At 90km is $O_2=1.47x10^{13}$ $cm^{-3}$, $M=7x10^{13}$ $cm^{-3}$, and $k1≈10^{-33}$ $cm^6 s^{-1}$ depending on temperature. The characteristic time has then an order of $10^6$ s. About one order smaller is the characteristic time of H, but still large. Both the production and the loss term of $HO_2$ depend on H and O being not in photochemical equilibrium in the lower thermosphere. Therefore a discrepancy relating to HO2 one should expect.*
Here, apparently, there was a misunderstanding caused by a possibly insufficiently clear indication in the manuscript. In these parts (Lines 304-318 (new 368-383), Section 6) we consider lower and middle mesosphere, heights below 0.01 mbar (~78km). In the revised manuscript, the upper height is 0.046 mbar (~71-72 km).

---

## Author Response (AR2)

Dear Editor,

Thank you very much for taking the time to handle the review process of our manuscript and providing useful recommendations!

With respect,

M. Kulikov, A. Nechaev, M. Belikovich, T. Ermakova, and A. Feigin

Response to the comments on the paper by Referee #1

Dear Referee,

We appreciate you taking the time to review our manuscript and we are grateful for your comments and constructive recommendations!

With respect,

M. Kulikov, A. Nechaev, M. Belikovich, T. Ermakova, and A. Feigin

**List of technical corrections:**

Below Referee's comments are in *blue italic*, our comments are in black.

1. *Line 38 "validate results of remote or in situ measurements". One should carefully consider the wording here. Models do not validate measurements. Perhaps "compare with results of measurements" is what is meant.*

Corrected. "validate" was replaced with "evaluate". See line 38 below.

2. *Line 193 "COMMA-IAP middle atmosphere dynamics" needs some explanation. This is also confusing because the previous paragraph states that the model dynamics is from CMAM.*

To avoid the confusion we deleted the sentence about COMMA-IAP model. See lines 194-196. The citations from the sentence were moved to the first sentence in Section 2. See lines 163-165.

3. *Line 241 "dashed area" looks like a shaded area or gray area in the figure.*

Corrected. "dashed area" was replaced by "gray area". As it is indeed looks like gray area in print. See line 242.

4. The acknowledgments were changed. See lines 432-434. The references were corrected. See lines 448, 462, 464, 499, 622, 736-738, 754-756, 778-783.

[revised manuscript text omitted]